# Region-specific laboratory reference intervals are important: A systematic review of the data from Africa

**Matt A. Price**[1,2]*, **Patricia E. Fast**[1,3], **Mercy Mshai**[4], **Maureen Lambrick**[5], **Yvonne Wangũi Machira**[4], **Lisa Gieber**[1], **Paramesh Chetty**[6], **Vincent Muturi-Kioi**[4]

1 IAVI, New York City, New York, United States of America, 2 Department of Epidemiology and Biostatistics, University of California at San Francisco, San Francisco, California, United States of America, 3 Division of Infectious Diseases, Stanford University School of Medicine, Palo Alto, California, United States of America, 4 IAVI, Nairobi, Kenya, 5 Laboratory Consultant, Cape Town, South Africa, 6 IAVI, Johannesburg, South Africa

* mprice@iavi.org

**Data Availability Statement:** This is a systematic review. The source data (i.e., data abstracted from reviewed manuscripts) are in an Excel file in the Supporting Information files.

## Abstract

Region-specific laboratory reference intervals (RIs) are important for clinical trials and these data are often sparse in priority areas for research, including Africa. We reviewed data on RIs from Africa to identify gaps in the literature with a systematic review of PubMed for RI studies from Africa published ≥2010. Search focus included clinical analytic chemistry, hematology, immunological parameters and RIs. Data from adults, adolescents, children, pregnant women, and the elderly were included. We excluded manuscripts reporting data from persons with conditions that might preclude clinical trial participation in studies enrolling healthy volunteers. Of 179 identified manuscripts, 80 were included in this review, covering 20 countries with the largest number of studies in Ethiopia (n = 23, 29%). Most studies considered healthy, nonpregnant adults (n = 55, 69%). Nine (11%) studies included pregnant women, 13 (16%) included adolescents and 22 (28%) included children. Recruitment, screening, enrollment procedures and definition of age strata varied across studies. The most common type of RIs reported were hematology (66, 83%); 14 studies (18%) included flow cytometry and/or T cell counts. Other common tests or panels included liver function assays (32, 40%), renal function assays (30, 38%), lipid chemistries (17, 21%) and serum electrolytes (17, 21%). The number of parameters characterized ranged from only one (three studies characterized either CD4+ counts, D-dimer, or hemoglobin), to as many as 40. Statistical methods for calculating RIs varied. 56 (70%) studies compared their results to international RI databases. Though most presented their data side-by-side with international data with little accompanying analysis, nearly all reported deviation from comparator RI data, sometimes with half or more of otherwise healthy participants having an "out of range" result. We found there is limited local RI data available in sub-Saharan Africa. Studies to fill this gap are warranted, including efforts to standardize statistical methods to derive RIs, methods to compare with other RIs, and improve representative participant selection.

**Funding:** This IAVI study was entirely funded by The Bill & Melinda Gates Foundation (reference number INV-024949). No author received any funding from any other source. The funder played no role in the data collection, analysis or writing of the manuscript, or the decision to submit it for publication.

**Competing interests:** The authors have declared that no competing interests exist.

## Introduction

Laboratory reference intervals (RIs) for a laboratory test are defined as that interval within which falls 95% of the healthy population. Accurate and appropriate RIs are important for the interpretation of clinical laboratory data, and are vital to guide the design, conduct and interpretation of clinical trials to ensure volunteer and product safety. Since 1992, systematic guidelines on the creation of reference intervals have been available from the Clinical Laboratory Standards Institute (CLSI) and the International Federation of Clinical Chemistry (IFCC) [1] and more recently new methodology has been proposed as part of a global effort under the IFCC Committee on Reference Intervals and Decision Limits (C-RIDL) to standardize generation of RIs [2–4]. Because establishing RIs is expensive, logistically demanding, and time-consuming, many laboratories rely on RIs from assay package inserts, textbooks, or published data, which may be derived from very different populations. However, international studies employing systematic methodologies have demonstrated important differences between populations for some analytes [5, 6].

Other issues hamper the interpretation of regional RIs. Research Investigators often employ convenience sampling methods, enrolling blood donors, institute staff or students for their studies of reference intervals. While often easier and less expensive, this could introduce a selection bias if the population studied is systematically different from the population against which the RIs are meant to be used (e.g., younger, different patterns in alcohol and/or tobacco consumption, not pregnant, or living with HIV). Selection of study participants should be documented and should include factors such as age, sex, evaluation of health (e.g., at a minimum with a questionnaire, ideally with a physical examination and laboratory testing for pathological conditions) so that partitions of meaningful subclasses may be well described [1]. Additional methodology problems include insufficient sample size, poorly defined statistical methods, incomplete or non-standardized data collection (e.g., demographics, medical history), and inconsistent screening and enrollment procedures.

Sub-Saharan Africa is increasingly being recognized as an important region to conduct clinical trials [7, 8], and thus appropriate RIs are needed to manage volunteer wellbeing and guide the interpretation of clinical trial results. Between 2004 and 2006 IAVI conducted a large, multisite study (including teams from Kenya, Uganda, Rwanda and Zambia) to establish RIs among persons potentially suitable for participation in HIV prevention trials [5, 9]. Since then, large-scale, systematic efforts to characterize regional RIs in India, China and elsewhere around the world have been published [2, 6, 10, 11] but data from sub-Saharan Africa has been more limited. Here we review publications presenting the generation of newer RIs specific to sub-Saharan Africa.

## Methods

A systematic review of Pubmed and Pubmed Central was conducted in November 2020 to search for laboratory reference intervals in sub-Saharan African populations published in 2010 and later. The search was repeated in October 2021 to update any new publications in the interim. IAVI's laboratory RI study included work done in 2004–6 [5, 9]; in the interest of focusing on more recent work, we excluded work conducted prior to 2006 (regardless of publication date). The review adhered to the Preferred Reporting Items for Systematic Reviews and Meta-Analyses (PRISMA) guidelines (http://www.prisma-statement.org/, accessed 7 January 2022). We did not pool data for any type of meta-analyses, as study designs, source populations, the nature of the data, and the rationale for this review made this irrelevant. The search focus included concepts related to standards in clinical, analytic chemistry, hematology, metabolism, immunological and other assays or markers of important substances found in

blood, urine, tissues, and other human biological fluids and reference values/intervals, i.e., the range or frequency distribution of a measurement in a population that has not been selected for the presence of disease or abnormality. Manuscripts that considered only clinical decision limits (i.e., thresholds or ranges of analytes associated with disease or negative clinical outcomes) were not considered. We limited our initial search to studies conducted in Africa. Highly relevant papers were also reviewed for citations and cited papers and with the Pubmed feature *find similar articles*. Relevant manuscripts that were identified were checked for any additional manuscripts not initially identified. This review was not registered, and a protocol was not prepared. The following terms were searched as MESH terms and as free text words:

blood proteins/standards, c reactive protein/standards, blood sedimentation/standards, ferritins/standards, cytokines/standards, chemokines/standards, vitamin d/standards, blood cell count/standards, clinical laboratory techniques/standards, hematology/standards, reference standards/standards, reference values/standards, bilirubin/standards, metabolism/standards, eosinophils/metabolism, hemoglobins/standards, immunoglobulin g/standards, l lactate dehydrogenase/standards, neutrophils/methods, allergy and immunology/standards, medical laboratory science/standards, blood proteins/standards, blood chemical analysis/standards, clinical chemistry tests/standards, reference intervals.

A secondary search for "African reference standards" and synonyms was performed in African Journal Online (https://www.ajol.info/index.php/ajol), Sabinet (https://www.sabinet.co.za/), Google Scholar, Dimensions (https://www.digital-science.com/products/dimensions/), and Project Smile (https://resources.psmile.org/resources/equipment/reference-intervals/reference-ranges). Additional manuscripts were identified by review of selected publication references or by notification from an author.

One author (MP) reviewed each manuscript to extract details including year(s) of study, country or region of study, sample size, study population (i.e., healthy adults, pregnant women, adolescents [typically ages 13–17], children [typically 12 and under and age-stratified], or other), RI calculation methods, and analytes included. Then two authors (VK, PF) reviewed the abstracts independently and, where necessary, the extracted details to further screen manuscripts; both VK and PF had access to all papers. Studies were excluded if they were too old (study conduct was prior to 2006), RIs were not calculated or were calculated for a group with a specific pathological condition or environmental exposure that may not be broadly relevant for clinical trial participants in sub-Saharan Africa (we did seek studies of healthy persons living with HIV, though we did not find any), or the study was not conducted in sub-Saharan Africa. Some manuscripts did not specify when the study was conducted; we reviewed the manuscript for context about timing and opted to include these studies, as it did not appear that they were done prior to 2006. Reviews and editorials were also excluded, though their references were checked for additional manuscripts missed in our initial searches.

Data presented here include year(s) of study; country or regions of study; study population size, recruitment methods, and composition (including ages and methods to describe "health"); methods used to generate RIs; number, type, and category of analytes included; and whether comparisons were made with international findings and how these comparisons were presented (e.g., were they quantified by presenting the number of 'out of range' values when compared against another RI range? Were statistical tools employed to test reported RIs against existing ranges?). Bias was assessed by a narrative evaluation of selected study populations and statistical methods. Analytes are listed individually and broadly grouped into 8 categories in Table 1, including hematology, immunology/ lymphocyte subsets, liver/pancreas function, kidney function, blood gas parameters, serum electrolytes, lipid parameters, and

**Table 1. Categorization of reported parameters in 80 manuscripts of RIs in Africa.**

| Parameter family | Parameters measured | Number (%) of studies with these parameters |
|---|---|---|
| Hematology Parameters | Total white cell count (including either three or five part differential), hemoglobin, platelet count, platelet distribution width, platelet larger cell ratio, plateletcrit, mean corpuscular volume, mean corpuscular hemoglobin concentration, mean corpuscular hemoglobin, red blood cell count, RBC distribution width by standard deviation, hematocrit, serum iron, unsaturated and total iron binding capacity, serum transferrin, percent transferrin saturation, activated partial thromboplastin time, prothrombin time, D-dimer, ferritin, mean platelet volume, reticulocytes | 66 (83%) |
| Liver and Pancreas Function Chemistry Parameters | Aspartate amino transferase, alanine amino transferase, alkaline phosphatase, albumin, total protein, gamma (γ) glutamyl transferase, total bilirubin, direct bilirubin, lipase, amylase, lactate dehydrogenase | 32 (40%) |
| Renal Function Chemistry Parameters | Blood urea nitrogen, serum cystatin C, creatinine, uric acid | 30 (38%) |
| Immunological/ Lymphocyte Subset Parameters | Immunoglobulins (e.g., IgM, IgG, IgA), complement component 3, lymphocyte subsets (e.g., CD4+, CD8+, CD3+, CD3+/CD4+, CD3+/CD8+, CD3-/CD56+, CD16+/56+, CD3+/HLA DR+, CD3+/CD4+/HLA DR+, CD3+/CD8+/HLA DR+, CD3+/CD4+/45RA+, CD3+/CD4+/45RO+, CD3+/CD8+/45RA+, CD3+/CD8+/45RO+, CD19+) | 23 (29%) |
| Lipid Metabolism Parameters | Triglycerides, cholesterol (total, HDL, LDL), | 17 (21%) |
| Serum Electrolyte Parameters | Bicarbonate, sodium, potassium, chloride, phosphate (Inorganic phosphorus), calcium, magnesium | 17 (21%) |
| Blood Gas Analysis Parameters | pH, Partial pressure of oxygen, carbon dioxide | 5 (6%) |
| Other Chemistry Parameters / Markers | Creatine kinase, blood glucose/fasting blood glucose, prostate-specific antigen, uric acid, C-Reactive Protein, thyroglobulin, thyroid stimulating hormone, free thyroxine, free triiodothyronine, folate, vitamin B12, lactate to pyruvate ratio, Cholinesterase, Free T4 (thyroxine), Free T3 (tri-iodothyronine) | 15 (19%) |

others (including thyroid tests, diabetes, tumor markers, markers of inflammation, etc.). A list of abbreviations is shown in Table 2. Text on participant health and RI comparisons shown in Table 3 are often copied or paraphrased directly from the source manuscript.

## Results

### Populations studied

2,383 manuscripts were screened to identify 179 manuscripts (Including 5 from reviewing reference sections or by word-of-mouth; Fig 1) of potential interest; 80 were included in this review, covering 20 countries across sub–Saharan Africa (Fig 2) with the largest number of studies in Ethiopia (n = 23, 29%); only one study included more than one country (Malawi and Uganda).

Most studies considered healthy, nonpregnant adults (n = 55, 69%). Nine (11%) studies included pregnant women, 13 (16%) included adolescents (ages 11–19; one study included 20 year olds) and 22 (28%) included children (<14 years old, typically stratified by age); three studies included elderly participants (4%) though specific age ranges varied and in one case the age range was not reported (Table 3). "Adult age" was not consistently defined, with 8 papers focused on "adult" RIs that included adolescents as adults, ranging in age starting from 13–17, and five papers started adult enrollment with volunteers over 18, with the lower age limit ranging from 19–25 years old; 6 studies also did not report either the upper and/or lower limit for adult ages (typically reporting median ages, Table 3). Several papers (10, 13%) considered multiple age groups (e.g., infants, children and adolescents were common for papers addressing RIs in the context of malaria prevention and control). Data for infants, children and adolescents were all age-stratified, though the age strata were not consistent across studies. Data for adolescents were typically also gender-stratified, but not always.

**Table 2. List of abbreviations.**

| | |
|---|---|
| 2HPP | 2-hour post-glucose plasma glucose (2HPP) |
| ALB | Albumin |
| ALP | alkaline phosphatase |
| ALT | alanine aminotransferase |
| aPTT | activated partial thromboplastin time |
| AST | aspartate amino transferase |
| BUN | blood urea nitrogen |
| C3 | complement component 3 |
| CBC | complete blood count |
| CK | creatine kinase |
| Cl | chloride |
| CO2 | carbon dioxide |
| Cr | creatinine |
| CRP | c-reactive protein |
| FBS | fasting blood sugar |
| Fe | iron |
| GGT | gamma-glutamyl transferase ($\gamma$-GTP) |
| Hb | hemoglobin |
| HCO3 | bicarbonate |
| Hct | hematocrit |
| HDL | high density lipoprotein cholesterol |
| K | potassium |
| LDH | lactate dehydrogenase |
| LDL | low-density lipoprotein cholesterol |
| MCH | mean corpuscular hemoglobin |
| MCHC | mean corpuscular hemoglobin concentration |
| MCV | mean cell volume |
| MID | basophils, eosinophils, and monocytes |
| MPV | mean platelet volume |
| Na | sodium |
| NK | natural killer cells |
| NRBC | nucleated red blood cell count |
| PCT | plateletcrit |
| PCV | packed cell volume |
| PDW | platelet distribution width |
| P-LCR | platelet larger cell ratio |
| PT | prothrombin time |
| RBC | red blood cells / erythrocytes |
| RDW | red cell distribution width |
| TF | transferrin |
| TG | triglycerides |
| U | Urea (BUN) |
| UA | uric acid |
| VCT | HIV counseling and testing (i.e., HCT) |
| WBC | total white blood cell count / leukocytes |
| $\gamma$-GTP | $\gamma$-glutamyl-transferase (GGT) |

**Table 3. Listing of 80 manuscripts of RIs in Africa including details on year(s) of study, size, population and recruitment, parameters measured (see Table 2 for abbreviations), and any reported comparisons with other RIs.**

| References | Years of study | Country/ies of study | N | Study population, recruitment, and type* | Ages** | Definition of "healthy"*** | Parameters measured | RI Methods**** | Analysis platforms | Comparisons across RI data sets / data sources |
|---|---|---|---|---|---|---|---|---|---|---|
| Kone 2017 [18] | 2004–2013 | Mali | 213 | Adults recruited at University Teaching Hospital, limited description of recruitment | 18–59 | We defined a healthy volunteer, as a participant with no clinical evidence of TB or HIV and no physical symptom of illness for more than 2 weeks before enrollment, including no evidence of fever, or weight loss. **No laboratory screening tests reported** | WBC, RBC, Hb, Hct, MCV, MCH, MCHC, RDW, Plt, MPV, Neutrophils, Lymphocytes, Monocytes, Eosinophils, Basophils, CD4 +, CD8+ T cells | Other | Coulter counter analyzer (Coulter Ac T diff, Beckman Coulter, Miami, FL); FacsCalibur (FASCalibur, BD, Biosciences, San Jose, CA, USA) | The authors reported that the hematological parameters' ranges were mostly different to the universal established ranges (instrument-provided RIs) and presented in a table against values from Togo, Burkina Faso, The Gambia, Nigeria, Mozambique, Tanzania, Ethiopia, and Uganda. No statistical testing was reported. Comparisons were reported in a table, differences were not quantified. |
| Alemnji 2010 [19] | 2006 | Cameroon | 501 | Adults from urban and rural areas responding to call for VCT (rural) and blood donors (urban) | 18–59 | Participants with the following outcomes that could affect the serum biochemical parameters were excluded from the study (75 out of the 576 initial participants): pregnancy, **HIV-positive status, hepatitis B/C infection, malaria,** and urine glucose | AST, ALT, ALP, Creatinine, Total Protein, Albumin, Triglyceride, HDL Cholesterol, LDL Cholesterol, Total Cholesterol, Total Bilirubin, Direct Bilirubin | Other | Not reported | The authors present their results compared to those from 5 clinical laboratories in Cameroon. The authors describe these RIs as "based on values obtained in various kits standardized on foreign populations." They also note their methods were comparable to a study done in Rwanda, but do not present specifics regarding differences. No statistical testing was reported. Comparisons were reported in a table (except the data from Rwanda, which was not presented), differences were not quantified. |
| Kumwenda 2012 [20] | 2006–2007 | Malawi & Uganda | 541 | Infants consecutively identified at the time of birth before discharge of mothers from the hospital or at postnatal maternal–child health clinics | 0-6m | Healthy, full-term newborns weighing >2.5 kg, **HIV** negative mothers | CD4+, CD8+ T cells, Eosinophil, Lymphocyte, Monocyte, Neutrophil, Basophil, White blood cells, Red blood cells, Platelets, Hemoglobin, Hematocrit, Sodium, Potassium, GGT, Creatinine, CO2, Chloride, BUN, Bilirubin total, AST, ALT, ALP | Other | Beckman–Coulter AcT5 Part Diff (California, USA); Malawi only: Beckman–Coulter CX5 Chemistry analyser (Brea, California, USA); Uganda only: COBAS Integra 400+ (Roche Diagnostics, Indianapolis, IN, USA) | The authors used the USA-published text book Harriet Lane Handbook: A Manual for Pediatric House Officers as the reference intervals for comparison. For example, mean haemoglobin and haematocrit values of Malawian and Ugandan infants in this study were consistently lower than those of Caucasian infants after the first week of life. The authors refer to another study where these results are presented against the DAIDS toxicity tables (but no RIs were presented) and many values were found to be out of range. No statistical testing was reported. Comparisons were reported in a table, differences were not quantified. |

(*Continued*)

**Table 3.** (Continued)

| References | Years of study | Country/ies of study | N | Study population, recruitment, and type* | Ages** | Definition of "healthy"*** | Parameters measured | RI Methods**** | Analysis platforms | Comparisons across RI data sets / data sources |
|---|---|---|---|---|---|---|---|---|---|---|
| Buchanan 2010 [21] | 2006–2008 | Tanzania | 655 | Children and adolescents enrolled from vaccination, ANC and other clinics | birth–11m, 1–4, 5–12, 13–17 | Healthy Tanzanian children/ adolescents, **HIV negative**, with no clinically apparent acute or chronic illness, were not currently on any medications, or were pregnant | Hemoglobin, Hematocrit, MCV, Platelets, WBC, RBC, Lymphocyte, Neutrophils, Monocytes, Eosinophils, Basophils, CD4+, CD8+ T cells | CLSI | Beckman Coulter AcT 5 Diff (Beckman Coulter, Fullerton, CA, USA); FACSCalibur (Becton Dickinson Biosciences, San Jose, CA, USA) | The authors compare their results to Ugandan data, and data from the USA and Europe. The authors describe differences across many parameters between their RIs and the comparator RIs. In the discussion, the authors also apply the U.S. National Institute of Health Division of AIDS (DAIDS) adverse event grading criteria commonly used in clinical trials to their RIs. They found that 128 (21%) of 619 children would be classified as having an adverse event related to Hb level, with 10% grade 1 (mild), 7% grade 2 (moderate), and 3% grade 3 (severe), yet 88% of them would be within this Tanzanian normal 95% reference interval. Further, 13% of all the children in our study would be classified as having an ANC related event. Even more notable, 29 (27%) of 109 healthy infants <12 months would be classified as having an ANC-related toxicity, with all but three being within the normal 95% reference interval for this population. No statistical testing was reported. Comparisons were reported in a table, differences between Ugandan, US, European data were not quantified, out of range values were calculated against DAIDS criteria. |
| Humberg 2011 [22] | 2006–2008 | Gabon | 226 | Infants and children screened for malaria vaccine trial | 4–9w, 18–60m | Healthy children; excluded: acute or serious chronic disease, recent use of investigational or nonregistered drugs, administration of immunoglobulins, blood transfusions within 3 months preceding the blood sampling, administration of immunosuppressants within 6 months prior to blood sampling, same sex twin, a family history of congenital or hereditary immunodeficiency, history of splenectomy, major congenital defects, weight-for-age z-score <-2, systemic infections, diarrhea, otitis media, antibiotic treatment up to 8 days before blood sampling, burns and large abscesses. Mild skin diseases or mild infections (e.g. rhinitis, conjunctivitis) were not exclusion criteria. **No laboratory screening tests reported** | RBC, Hb, Hct, MCV, MCH, MCHC, PLT, WBC, LYM, MON, NEU, EOS, BAS, GPT, Cr | CLSI | ABX Pentra 60 analyser; ABX MIRA PLUS analyser | The authors compared their results to those from two European textbooks from 1998 and 2005, published in Germany. Compared to European populations, values for several red cell parameters (hemoglobin, hematocrit, red blood cell count, mean corpuscular volume) were lower and platelet counts were higher. Eosinophils were higher in the older age group, most likely caused by intestinal helminths. No statistical testing was reported. Comparisons were reported in a table, differences were not quantified. |

(*Continued*)

**Table 3.** (Continued)

| References | Years of study | Country/ies of study | N | Study population, recruitment, and type* | Ages** | Definition of "healthy"*** | Parameters measured | RI Methods **** | Analysis platforms | Comparisons across RI data sets / data sources |
|---|---|---|---|---|---|---|---|---|---|---|
| Odutola 2014 [23] | 2006–2010 | The Gambia | 675 | Infants of mothers attending Infant Welfare Clinic of periurban Health Centre were invited to participate | 3–10m | Apparently healthy children with no clinically significant acute or chronic illness (e.g., no malaria or diarrhea in past week), **samples checked for parasites.** Included small number (17) of wasted children | Hemoglobin, red blood cell count, white blood cell count, neutrophils, lymphocytes, monocytes, platelets, mean corpuscular volume, Potassium, Chloride, Urea, Creatinine, AST, ALT, ALP, Bilirubin | Other | CELL-DYN 3700 sample loader (ABBOTT, USA); VITROS 350 Analyzer (Ortho Clinical Diagnostics, USA) | The authors compared their RIs to those from two textbooks (published in the USA), and compare to data from Nigeria and Tanzania in very broad terms in the discussion section. Their set of hematological and biochemical reference values for healthy infants in The Gambia differs from values in other settings. No statistical testing was reported. Comparisons were reported in a table, differences were not quantified. |
| Gitaka 2017 [24] | 2007 | Kenya | 1493 | Children enrolled in RTS,S clinical trials | 4w–7m | No serious acute or chronic illness as determined by history and physical examination (defined as not having any signs and symptoms of disease, ambulatory (children older than 1 year) and not underweight, defined as weight-for age Z score (WAZ) -2), medical history records or laboratory screening tests were eligible, no history of allergic reactions, previous blood transfusion, major congenital defects, or HIV disease Stage III or Stage IV. **No laboratory screening tests reported.** | Hb, Hct, MCHC, MCV, platelets, WBC, neutrophils, lymphocytes, monocytes, eosinophils, basophils, ALT, Cr | CLSI | Beckman Coulter AcT 5 Diff Haematology Analyzer (Beckman Coulter, USA); Vitalab Selectra-E clinical chemistry analyser (Vital Scientific (Merck), Netherlands) | Findings were compared with published ranges from Tanzania, Europe and The United States. reference ranges in infants largely overlapped with those from United States or Europe, except for the lower limit for Hb, Hct and platelets (lower); and upper limit for platelets (higher) and hematocrit (lower). Community norms for common hematological and biochemical parameters differ from developed countries. No statistical testing was reported. Selected comparisons were reported in a table (not all RIs were shown), differences were not quantified. |
| Segolodi 2014 [25] | 2007–2010 | Botswana | 1786 | Adults screened for TDF2 study | 18–39 | Healthy Botswana adults, **limited definition of "health"**, excluded positive for **HBsAg, HIV-**positive, on medication for chronic illness and pregnant/ breastfeeding. | Hb, Hct, creatinine, inorganic phosphorus, HCO3, potassium, sodium, chloride, BUN, direct bilirubin, total bilirubin, serum amylase, AST, ALT | CLSI and other | A Sysmex XT1800i hematology analyzer (Sysmex, Kobe, Japan); Roche Integra 400plus analyzer | The authors compare to BOTUSA (western derived ranges), DAIDS Toxicity grading tables, US RIs, and RIs from eastern and southern Africa. BOTUSA reference ranges would have classified participants as out of range for some analytes, with amylase (50.8%) and creatinine (32.0%) producing the highest out of range values. Applying the DAIDS toxicity grading system to the values would have resulted in 45 (2.5%) and 18 (1%) participants as having severe or life-threatening values for amylase and hemoglobin, respectively. No statistical testing was reported. Comparisons were reported in a table and text. |

**Table 3.** (Continued)

| References | Years of study | Country/ies of study | N | Study population, recruitment, and type* | Ages** | Definition of "healthy"*** | Parameters measured | RI Methods**** | Analysis platforms | Comparisons across RI data sets / data sources |
|---|---|---|---|---|---|---|---|---|---|---|
| Odhiambo 2015 [26] | 2007–2010 | Kenya | 953 | Adolescents and adults, sexually active and screened for observational cohort study to estimate HIV incidence | 16–17, 18–34 | Excluded **HIV**, pregnancy. Tested for **HSV-2 and other STI** (data not reported). Enrollees from in an observational prospective cohort study known as the Kisumu Incidence Cohort Study. Had to be "healthy" (**not defined**) | WBC, neutrophils, lymphocytes, monocytes, eosinophils, basophils, RBC, Hb, Hct, MCV, MCH, platelet counts, ALT, Cr, BUN | CLSI | Coulter ACT 5Diff CP analyzer (Beckman Coulter, France); Cobas Integra 400 plus biochemistry analyzer (Roche, Germany) | The authors compared their RIs to published region-specific reference intervals (western Kenya and the USA) and the 2004 NIH DAIDS toxicity tables. They found that while 10% would be if using the US DAIDS. Blood urea nitrogen was most often out of range if US based intervals were used: <10% abnormal by local intervals compared to 83% (men) and 95% (women) by US based reference intervals. No statistical testing was reported. Comparisons were reported in a table and text. |
| Mine 2012 [27] | 2008 | Botswana | 261 | Adults recruited from HIV testing centers around the capital city | 18–66 | Apparently healthy **HIV-negative volunteers**, excluded via screening questionnaire: pregnant, breastfeeding, inpatients in a hospital or subjectively ill during the last month, rec'd medical treatment (no time specified), recent or recurrent infection, including HIV and malaria, smoked in the hour before blood was drawn, donated blood in the preceding month | WBC, lymphocytes, monocytes, eosinophils, neutrophils, basophils, a 14-parameter hemogram, integral reticulocyte analysis that includes an immature reticulocyte fraction, NRBC, MCV, MCH, MCHC, RDW-SD | CLSI | FACSCalibur flow cytometer (Becton Dickinson Immunocytometry Systems, San Jose, California); Sysmex XE-2100 (Sysmex, Kobe, Japan) | The authors compare their results to those from blood donors also from Botswana, as well as volunteers from eastern and southern Africa (Kenya, Rwanda, Uganda and Zambia), Ethiopia, Uganda, and the USA. Their hemoglobin reference intervals were generally higher than in East Africa, but lower than those from Ethiopia and US-based comparison populations. No statistical testing was reported. Comparisons were reported in a table, differences were not quantified. |
| Kueviakoe 2011 [28] | 2008 | Togo | 1349 | Adult blood donors, no description of recruitment. | 17–58 | Being "healthy" was not explicitly reported as a requirement in the methods. **Negativity of all serological tests for viral infections (not specified) and for malaria**, normal hemoglobin electrophoresis screening, and absence of hypochromia on peripheral blood smear. | WBC, RBC, hemoglobin, hematocrit, MCV, MCH, MCHC, platelets, neutrophils, lymphocytes, eosinophils, basophils, monocytic cells | Other | Sysmex SF-3000 automated hematology analyzer | The authors compared their RIs to those from Uganda, Central African Republic, Ethiopia, South Africa, Gambia, Kenya, and Tanzania. The median values are similar to the ones found in other studies performed in Africa and even in northern countries (data are not shown). The ranges of different parameters are often different especially for RBC considering the influence of alpha-thalassemia and iron deficiency. The reason for lower absolute neutrophil count is still unclear. No statistical testing was reported. Comparisons were reported in a table, differences were not quantified. |

*(Continued)*

**Table 3.** (Continued)

| References | Years of study | Country/ies of study | N | Study population, recruitment, and type* | Ages** | Definition of "healthy"*** | Parameters measured | RI Methods **** | Analysis platforms | Comparisons across RI data sets / data sources |
|---|---|---|---|---|---|---|---|---|---|---|
| Woldemichael 2012 [29] | 2008–2009 | Ethiopia | 1965 | 1965 adults randomly selected from population based survey of NCD (same as Haileamlak 2012). | 15–64 | Included all apparently healthy individuals (not defined); disabled and acutely ill during the data collection were excluded. **No laboratory screening tests reported** | Total cholesterol, triglycerides, total serum protein, blood urea nitrogen, creatinine, uric acid, alanine aminotransferase, aspartate aminotransferase | Other | Human star 80 (Gesellschaft fur Biochemica und Diagnostica, Germany) | Presented in the discussion, the authors compare their findings to RIs from Cameroon, East and Southern Africa, and a textbook (based on western values). No statistical testing was reported. Comparisons were reported in a table, differences were not quantified. |
| Haileamlak 2012 [30] | 2008–2009 | Ethiopia | 1965 | 1965 adults randomly selected from population based survey of non-communicable diseases (same as Woldemichael 2012). | 15–64 | Included all apparently healthy individuals (not defined); disabled and acutely ill during the data collection were excluded. **No laboratory screening tests reported** | Hb, Hct, RBC, MCV, MCH, MCHC, WBC count (including neutrophils, mixed white cells, lymphocytes), platelet count, CD3+, CD4+, CD8+ T cells | Other | KX-21 hematology analyzer, (Sysmex Coporation, Germany); FACS system (Becton Dickinson San Jose California, USA). | Presented with limited descriptions in the discussion, the authors compare their results to RIs from Turkey, Uganda, Malaysia and the textbook Harrisons Principles of Internal Medicine (published in the USA). No statistical testing was reported. Comparisons were reported in a table, differences were not quantified. |
| Santana-Morales 2013 [31] | 2008–2010 | Ethiopia | 454 | Adults and children attending health screening at rural hospital, limited description of recruitment. *An additional 117 persons with malaria described in this paper were not considered for this review.* | 1–5, 6–12, 13–98 | The following categories were excluded: pregnant, **malaria infected (except comparison group), HIV-positive and intestinal helminth**-positive. | Hemoglobin | CLSI | Hemo-Control EKF Diagnostic Analyser | The RIs for the Gambo General Rural Hospital population in this study were compared with parameters obtained from other African populations and with parameters set by WHO. In general, the lower limits for adult hemoglobin range obtained from this population were slightly higher than those derived from Kenya, Uganda and Gambia, but were equal to those established by another study in Ethiopia and the hemoglobin range established by WHO (2011). In the child populations, the minimum values were lower than those obtained from different African populations and those established by WHO. No statistical testing was reported. Comparisons were reported in a table, differences were not quantified. |

*(Continued)*

**Table 3.** (Continued)

| References | Years of study | Country/ies of study | N | Study population, recruitment, and type* | Ages** | Definition of "healthy"*** | Parameters measured | RI Methods**** | Analysis platforms | Comparisons across RI data sets / data sources |
|---|---|---|---|---|---|---|---|---|---|---|
| Schmidt 2018 [32] | 2008–2012 | South Africa | 634 | Infants recruited for TB clinical trials from general population using vaccination clinic records, birth registers and referrals from community contacts and other participants. | 3-6m | BCG vaccinated at birth, HIV-unexposed, neither a maternal history of tuberculosis disease or exposure to infectious household contact, confirmed by either negative TST or IGRA. General good health & weight at the time of screening. **No laboratory screening reported.** | White blood cell count, Red blood cell count, Hemoglobin, Hematocrit, Mean corpuscular volume, Mean corpuscular hemoglobin, Mean corpuscular hemoglobin concentration, Platelet count, Total bilirubin, Alkaline phosphatase, Gamma-glutamyl transferase, Alanine transaminase, Aspartate transaminase | CLSI | Sysmex XN10 instrument (manufactured in Germany); Roche Cobas 6000 instrument (manufactured in Japan) | The authors calculated the percentage of observed values out of bound (in terms of lower and upper limits) compared to NHLS laboratory RIs (derived from Caucasian subjects) and found that parameters in apparently healthy South African infants deviate frequently from these RIs, including abnormalities consistent with subclinical hypochromic microcytic anemia. 91% of platelet values were higher than the NHLS RIs and over half, 55% and 57%, had MCV and MCH values below the lower limits of the NHLS RIs; 4 other RIs had significant (>10%) numbers of values out of range. No statistical testing was reported. Comparisons were reported in a table, figures, and text. |
| Gelaye 2011 [33] | 2009–2010 | Ethiopia | 1807 | Adult employees of the Commercial Bank of Ethiopia and teachers in government schools from the capital city. A multistage, probabilistic stratified random sampling strategy was used to identify and recruit participants. | 18–60 | subjects were excluded from these analyses if they were currently pregnant or on medication. Subjects older than 60 years of age were also excluded. **No laboratory screening tests reported** | WBC, RBC, hemoglobin, hematocrit, MCV, MCH, MCHC, platelets, neutrophils, lymphocytes, monocytes, eosinophils, basophils | CLSI | Not reported | No |
| Sagnia 2011 [34] | 2009–2010 | Cameroon | 352 | Children from Yaoundé township were recruited by healthcare assistants at the vaccination unit and by township social assistants. | birth-6y | A child was considered healthy when there was no history infectious diseases or hematological and immunological disorders. The parent or guardian of each participant completed a questionnaire. Participants were assessed for signs of febrile illnesses, active infection, malnutrition, and clinical AIDS. exclusion criteria were concurrent illness and/or medication, an axillary temperature of 38˚C, malnutrition (weight for height of <70%), an unknown date of birth, and known HIV/AIDS infection. Blood samples were rejected if they were **HIV** infected or **plasmodium** positive | Lymphocyte subset reference values CD3+, CD4+, CD8+, CD19+, NK, 4/DR/38, 4/DR, 4/38, 8/DR/38, 8/DR, 8/38, 3/4/RO, 3/4/RA, 4/RO, 4/RA, 3/8/RO, 3/8/RA, 8/RO, 8/RA, 4/RA/62L, 8/RA/62L | Other | FACSCalibur flow cytometer (BD Biosciences, San Jose, CA) | The authors compared their results to selected results from USA, The Netherlands, Malawi, Kenya, Saudi Arabia, Ethiopia, Zambia, and Uganda. Normal lymphocyte subsets values among children from Cameroon differ from reported values in Caucasian and some African populations. No statistical testing was reported. Comparisons were reported in a table, differences were not quantified. |

(Continued)

**Table 3.** (Continued)

| References | Years of study | Country/ies of study | N | Study population, recruitment, and type* | Ages** | Definition of "healthy"*** | Parameters measured | RI Methods**** | Analysis platforms | Comparisons across RI data sets / data sources |
|---|---|---|---|---|---|---|---|---|---|---|
| Dosoo 2014 [35] | 2009–2010 | Ghana | 1442 | Children and adolescents randomly selected from DSS. | 0.5–4, 5–12, 13–17 | General good health as determined by a clinician using medical history and physical examination, excluded adolescent pregnant females. Children with evidence of acute or chronic respiratory, cardiovascular, gastrointestinal, hepatic, or genitourinary conditions; history of blood donation/transfusion within three months preceding the survey; hospitalization within a month preceding the survey; or any other findings that in the opinion of the examining clinician may compromise the assessment of the laboratory parameters of interest in this study were excluded. **No laboratory screening tests reported.** | ALT, AST, Amylase, CK, GGT, LDH, total Protein, Albumin, total Bilirubin, Direct Bilirubin, Cholesterol, Glucose, Iron, Triglycerides, Urea, Creatinine, Uric acid, Chloride, Phosphorus, Potassium, Sodium, Hemoglobin, Hematocrit, RBC, MCV, MCH, MCHC, RDW, Platelets, PDW, Total WBC, Lymphocytes, Monocytes, Granulocytes | CLSI | ABX Micros 60 Hematology Analyzers (Horiba-ABX, Montpellier, France) | The authors compared their results to those from Kenya, Uganda, Tanzania, an unnamed "Western country", and lower limits provided from the WHO (presumably from "developed countries"). Hemoglobin, hematocrit, mean cell volume, erythrocytes, urea, and creatinine were lower when compared with values from northern countries but alanine aminotransferase, aspartate aminotransferase, and total bilirubin were higher. No statistical testing was reported. Comparisons were presented in a table; differences were not quantified. |
| Buchanan 2015 [36] | 2009–2011 | Tanzania | 619 | Infants, children and adolescents enrolled from schools and clinics; outreach at places of worship. | 1m–1y, 1–4, 5–12, 13–17 | No fever, no physical complaints, not ill on exam, no chronic illnesses, or meds, not pregnant, HIV negative | Albumin, alkaline phosphatase, ALT, AST, amylase, total bilirubin, calcium, cholesterol, creatinine kinase, bicarbonate, chloride, creatinine, glucose, HDL, LDL, potassium, lipase, phosphorus, magnesium, sodium, total protein, triglycerides, urea | CLSI and other | Cobas Integra 400 plus biochemistry analyzer (Roche Diagnostics, Germany) | The authors compared their data to the USA-published Harriet Lane Handbook and the US National Institutes of Health Division of AIDS (DAIDS) grading criteria for classification of adverse events. They found that for selected parameters, up to 15% or more of infants or children in certain age groups would have been categorized as having an adverse event as defined by DAIDS. No statistical testing was reported. Comparisons were presented in a table and quantified in the text. |
| Tembe 2014 [37] | 2009–2012 | Mozambique | 257 | Young adults participating in a study of the prevalence and incidence of sexually transmitted viruses at the youth-friendly clinic, Maputo Central Hospital. | 18–24 | Volunteers who were febrile, pregnant, or seropositive for **HIV, syphilis or hepatitis B surface antigens** were excluded from the study. No other description of "healthy" reported | WBC, RBC, platelets, lymphocytes, neutrophils, Hb, Hct, MCV, MCH, MCHC, RDW-SD, RDWCV, PDW, MPV, P-LCR, lymphocytes, neutrophils, monocytes, basophils, eosinophils, creatinine, total bilirubin, albumin, AST, ALT, glucose, urea, uric acid, amylase, HDL, cholesterol, triglycerides, ALP | CLSI and other | Sysmex KX-21N Hematology Analyzer (Sysmex Corporation, Kobe, Japan); Vitalab Selectra Junior (Vital Scientific, Dieren, Netherlands) | The authors compared their results to those from Uganda, Kenya, USA and the 2004 DAIDS toxicity tables. They found that their immunology ranges were comparable to those reported for the US and western Kenya. Hematologic values differed from the US values but were similar to reports of populations in western Kenya and Uganda. The chemistry values were comparable to US values, with few exceptions. However, as many as 69% would be ineligible for trial participation, when screened against the DAIDS tables. No statistical testing was reported. Comparisons were presented in a table and differences vs. DAIDS table were quantified. |

(*Continued*)

**Table 3.** (Continued)

| References | Years of study | Country/ies of study | N | Study population, recruitment, and type* | Ages** | Definition of "healthy"*** | Parameters measured | RI Methods**** | Analysis platforms | Comparisons across RI data sets / data sources |
|---|---|---|---|---|---|---|---|---|---|---|
| Ouma 2021 [38] | 2009–2014 | Kenya | 1,509 | 1509 (53% female) of 1631 Infants participating in GSK phase3 malaria vaccine trial which enrolled volunteers from a local Health and Demographic Surveillance System to ensure a representative sample (no details on the sampling strategy were reported) | 1m–17months | Data from a trained physician-conducted physical examination were reviewed as per the trial procedures to include healthy infants without acute or chronic, clinically significant pulmonary, cardiovascular, hepatobiliary, gastrointestinal, renal, neurological, mental or hematological functional abnormality or illness that required medical therapy. Data included medical history, physical examination (blood pressure, weight, pulse, Z score and vital signs) or clinical assessment before being enrolled into the study. Data from children who had recurrent infection, fever, including **HIV (or maternal exposure to HIV) and malaria** or severe anemia, defined as hemoglobin level of <5.0g/dl or hemoglobin concentration of <8.0g/dl associated with clinical signs of heart failure and/or severe respiratory distress or those receiving medical treatment at the time of sample collection were excluded from the study. | WBCs, RBCs, HGB, HCT, platelets, granulocytes, lymphocytes, and monocytes were directly measured. MCV, MCH, MCHC, RDW and MPV were extrapolated. The analyzer did not give separate counts of neutrophils, basophils and eosinophils. | CLSI | three-Part differential Coulter counter hematology analyzer | Includes comparisons with data from Kilifi (coastal Kenya), Tanzania, USA/Europe. The sources of data from Tanzania and USA/Europe were unclear, and no distinction was made between data from USA and Europe (unclear if data are combined, or which data are from which region). No statistical testing was reported. Comparisons were reported in a table, differences were not quantified. |
| Abera 2012 [39] | 2010 | Ethiopia | 405 | Adult attendees to VCT center in regional referral hospital. | 18–60 | **Negative for anti-HIV antibodies**, non-pregnant women, nonsmokers, Body Mass Index (BMI) ≥18.5 kg/m2, no history of current or recent morbid conditions such as gastrointestinal tract infections, active tuberculosis, no major surgery, no medication or allergy to drugs, no autoimmune diseases, and no history of blood transfusion | CD4+, CD8+ T cells, WBC, neutrophils, monocytes, eosinophils, basophils, some precursor cells, RBC, Hb, Hct, MCV, MCH, MCHC, Platelets | Other | FACS Count (Becton Dickinson, USA); Cell-DYN 1800 (Abbott Lab, USA). | Compared with accepted reference value of the National ART Laboratory [the source of which is not specified], 62 (21.4%), 30 (7.4%) and 36 (8.8%) of the participants had lower than the lower ranges of CD4+ T cells, platelets and leucocytes, respectively. The authors reported using ANOVA and chi squared tests to determine if out of range values differed by gender and found a statistically significant difference between males and females, where 21.5% males and 6.6% females had CD4+ T cells below the normal values (<500 cells/mm3) (P = 0.001). Table 3 shows comparison of the medians and means of the present study with results of earlier studies in Ethiopia, Kenya and Tanzania; data are presented in a table but no statistical testing is done to compare their ranges to these RIs. The authors recommend the National ART Laboratory guidelines be reevaluated for CD4+ T Cells, but that for hematology they remain acceptable. Comparisons are reported in a table and text. |

*(Continued)*

**Table 3.** (Continued)

| References | Years of study | Country/ies of study | N | Study population, recruitment, and type* | Ages** | Definition of "healthy"*** | Parameters measured | RI Methods**** | Analysis platforms | Comparisons across RI data sets / data sources |
|---|---|---|---|---|---|---|---|---|---|---|
| Okonkwo 2015 [40] | 2010 | Nigeria | 304 | Adults recruited using age and gender stratified random sampling to ensure that participants from each Local Government Council matched the overall population structure. | 20+, mean age 50 no upper limit given | Generally of good health with no evidence of a chronic or acute illness on clinical examination, excluded: Hypertension, diabetes mellitus, stroke, kidney disease, family history of kidney disease, significant proteinuria (urine dipstick protein ≥1+ equivalent to approx. 30 mg/dL), liver disease, and pregnancy. **No laboratory screening tests reported** | Serum Cystatin C, SCr, blood glucose | IFCC | Humalyzer 2000 Chemistry analyzer (Human GmBH Diagnostica Worldwide, Wiesbaden, Germany) | No |
| Palacpac 2014 [41] | 2010 | Uganda | 140 | Children, adolescents, and adults in a Phase Ib trial for malaria vaccine following outreach via radio and schools | 6–10, 11–15, 16–20, 21–32 | Healthy, with no obvious symptoms/signs of either acute or chronic respiratory, cardiovascular, gastrointestinal, hepatic or renal disease; no history of blood donation/ transfusion within one month and, for females, non-lactating and tested for pregnancy **(though not explicitly listed as exclusion criterion)**. Blood checked for **malaria (test not reported)**. | WBC, RBC, Hb, Hct, platelet count, ALP, ALT, AST, γ-GTP, amylase, albumin, total protein, total bilirubin, cholesterol, glucose, creatinine, BUN, uric acid, potassium, sodium | CLSI | Sysmex KX-21N (Sysmex Corporation, Kobe, Japan); Cobas C111 (Roche Diagnostics, Mannheim, Germany) | No |
| Melkie 2012 [42] | 2010–2011 | Ethiopia | 117 | Newborns and infants attending OBGYN and vaccination clinics, respectively, limited description of recruitment | newborns & infants | Newborns over 37 weeks of gestation, ≥2500 g birth weight and no history of fetal problems. Preterm newborns, newborns with <2500 g birth weight, newborns requiring intensive resuscitation and care, newborns from mothers with documented antenatal or intra-partum complications (gestational diabetes, **HIV, hepatitis B/C,** eclampsia, etc.), have organic diseases and other diseases that alter their biochemical profile, have taken medications that can affect their biochemical profile were excluded. Structured questionnaires were used in both cases for the collection of selected demographic information. | Albumin, AST, ALP, direct bilirubin, total bilirubin, ALT, GGT | CLSI | HumaStar 300 analyzer (Human diagnostics worldwide, Germany) | Upon comparison of the RIs obtained from this study with previously reported values of similar age group (assay package insert RIs, and two studies from Canada), visually significant differences were seen. The values were also different from respective adult values determined for adult Ethiopian population (see reference Table 8). No statistical testing was reported. Comparisons were reported in a table, differences were not quantified. |
| Samaneka 2016 [43] | 2011 | Zimbabwe | 769 | Adults, cross-sectional, representative multistage random sample of 3 regions (same study, different population as Gomani 2015) | 18–55 | Healthy, no history, clinical or laboratory evidence of chronic disease (hypertension, diabetes, liver, renal), **HIV and HBsAg** negative. No reported medications for acute illness (or were within one week past completion of a course of medication) with no abnormal findings on clinical and laboratory examination | White cell count, hemoglobin, automated five part differential, platelet count, MCV, MCHC, MCH, red blood cell count, hematocrit, sodium, potassium, chloride, phosphate, creatinine, creatinine kinase, calcium, urea, AST, ALT, ALP, albumin, total protein, GGT, total bilirubin, direct bilirubin, bicarbonate, triglycerides, lipase, cholesterol, HDL–cholesterol, LDL–cholesterol, glucose | CLSI | Sysmex XT2000I (Symex Europe D-22848Nordersted, Germany); Hitachi 902 (Hitachi, Roche diagnostics D-6829 2 Mannheim Germany) | No |

(Continued)

**Table 3.** (Continued)

| References | Years of study | Country/ies of study | N | Study population, recruitment, and type* | Ages** | Definition of "healthy"*** | Parameters measured | RI Methods**** | Analysis platforms | Comparisons across RI data sets / data sources |
|---|---|---|---|---|---|---|---|---|---|---|
| Gomani 2015 [44] | 2011 | Zimbabwe | 302 | Adolescents, community based, using multistage random sampling technique (same study, different population as Samaneka 2016) | 12–17 | Screened for HIV and Hepatitis B. Excluded those with chronic medical conditions like diabetes (no additional details provided). All participants had a physical examination performed by the study physician before enrollment into the study | White Cell, Red Cell, Hemoglobin, Hematocrit, MCV, MCH, MCHC, Platelets, Neutrophil, Lymphocyte, Monocytes, Eosinophil, Basophil, Red blood Cell distribution width, Urea, Sodium, Potassium, Chloride, AST, ALT, ALP, Amylase, Total Protein, Albumin, GGT, Total Bilirubin, Phosphate, Calcium, Bicarbonate, Triglycerides, Creatinine, Lipase, Cholesterol, LDL-C, HDL-C, Glucose, Direct Bilirubin | CLSI | Sysmex xt2000i (Sysmex Europe Norderstedt, Germany), Hitachi 902 (Hitachi, Roche Diagnostics Mannheim, Germany) | No |
| Beavogui 2020 [45] | 2011–2013 | Guinea | 450 | Children, adolescents and adults, no description of recruitment. Some enrolled for 3 visits, across seasons. | 6–10, 11–15, 16–45 | Apparently healthy clinically (this was not described), not pregnant. Everyone was tested for malaria, but test was not reported. | WBC, RBC, Hemoglobin, Hematocrit, MCV, MCH, MCHC, Platelet, MPV, Lymphocytes, Monocytes, Neutrophils, Eosinophils, Basophils, AST, ALT, Total Bilirubin, Creatinine | CLSI | ABX Pentra 60 Hematology Analyzers (Horiba-ABX, Montpellier, France); Piccolo® xpress™ Chemistry Analyzer (USA) | No |
| Odhiambo 2017 [46] | 2011–2014 | Kenya | 120 | Pregnant women, self-selected, seeking ANC care. Prospective/ follow up. | 14–44, no age strata | HIV uninfected, pregnant, with a plan to remain in the area until at least 9 months postpartum, willing to have serial visits at maternal child health clinic with serial HIV testing through 9 months postpartum, not on medication that could alter hematological parameters, and no clinical evidence of malaria and helminth infections | ALT, AST, Bilirubin, Cr, CD4 + T cells, CD8+ T cells, Hct, Hb, MCV, RBC, WBC, Neutrophils, Monocytes, Lymphocyte count, Basophils, Eosinophils, Platelets | CLSI | Coulter ACT 5Diff CP analyzer (Beckman Coulter, France) | The authors compared their results to non-pregnant values from Kenya and the US, as well as the 2004 DAIDS Toxicity Tables, using the Wilcoxon text. There were substantial differences in U.S. and Kenyan values for immune-hematological parameters among pregnant/postpartum women, specifically in red blood cell parameters in late pregnancy and 2 weeks postpartum. Using the new hemoglobin reference levels from this study to estimate prevalence of 'out of range' values in a prior Kisumu research cohort of pregnant/ postpartum women, resulted in 0% out of range values, in contrast to 96.3% using US non-pregnant reference values. Comparisons were reported in a table and the text. |
| Pennap 2011 [47] | Not reported | Nigeria | 444 | University students, no description of recruitment. | 15–44 | Participants who were either HIV-, HBV- or HVC-positive or who had apparent signs or symptoms of ill-health were excluded from the study. Volunteers who indicated that they were on immunomodulatory drugs or had a history of idiopathic CD4þ T-lymphocytopenia were also excluded. | CD4+ T cells | Other | BD FACScount cytometer | No |

(Continued)

**Table 3.** (Continued)

| References | Years of study | Country/ies of study | N | Study population, recruitment, and type* | Ages** | Definition of "healthy"*** | Parameters measured | RI Methods**** | Analysis platforms | Comparisons across RI data sets / data sources |
|---|---|---|---|---|---|---|---|---|---|---|
| Maphephu, 2011 [48] | Not reported | Botswana | 309 | Adult anonymized residual blood samples from blood donors. | 16–60 | A professional health counsellor carries out a pre-donation confidential assessment to determine eligibility to donate blood and prior history of diseases, medications, sexual behavior, cigarette smoking, alcohol consumption and illicit drugs. All the blood collected was tested for presence of **HIV and hepatitis A, B and C viruses, and screened for sexually transmitted infections.** | Total cholesterol, HDL cholesterol, non-HDL cholesterol | CLSI | COBAS Integra 400 | The authors compare the cut-of values of total cholesterol concentration in a table showing their Batswana results and those from North American populations according to recommendations of the Laboratory Standardization Panel of the National Cholesterol Education Program. No statistical testing was reported. Comparisons were reported in a table, differences were not quantified. |
| Amah-Tariah 2011 [49] | Not reported | Nigeria | 220 | Pregnant women first time attendees to ANC clinic. | 18–42 | Apparently healthy pregnant subjects. No subject had a history of hematinic or any other mineral supplementation during the present pregnancy prior to recruitment into the study. No subject had a history of vaginal bleeding during the present pregnancy and none had received any form of blood transfusion within the past eight months. **No laboratory screening tests reported.** | Serum iron, iron binding capacity, serum transferrin, percent transferrin saturation, red cell distribution width, platelet count, mean platelet volume, platelet distribution width, platelecrit, platelet large cell ratio | Other | Prestige 24i automated clinical analyzer (Tokyo Boeki Medical System Ltd. Tokyo, Japan) | No |
| Mugisha 2016 [50] | 2012–2013 | Uganda | 903 | Older adults in General Population Cohort study participating in anemia survey. Limited description of recruitment. | 50–64, 65+, 8% > = 80 | Participants were excluded if they were too ill to answer questions or couldn't remember their age. Tested for **malaria, hookworm, HIV** as exclusionary criteria. | Hemoglobin, Hematocrit, Erythrocytes, Platelets, WBC, MCH, MCV, MCHC, Neutrophils, Lymphocytes, Monocytes, Eosinophils, Basophils | CLSI | Beckman Coulter AC.T 5 diff CP Haematology analyzer (Beckman Coulter, USA) | Compared to the reference intervals from old people in high income countries (Australia, China and two studies from USA), all the hematology parameters from this study population were low, with the exception of MCV values compared to the study from China. No statistical testing was reported. Comparisons were reported in a table, differences were not quantified. |
| Okebe 2016 [51] | 2012–2013 | The Gambia | 1141 | Children randomly selected from DSS for 2 cross sectional surveys before and after rainy season. No attempts were made to exclude a child based on participation in the previous survey. | 12–59m | The medical history taken by the team nurse focused on any episode of illness such as fever, frequent watery stools, and antibiotic use in the preceding two weeks, blood transfusions or any known medical condition such as sickle cell disease. A brief physical examination comprised of axillary temperature measurement, weight and height measurements, auscultation of the chest and abdominal palpation for enlarged spleen or liver was also done. Only fever was explicitly mentioned as exclusion criteria, febrile recruits (only) were screened for malaria by RDT. **No laboratory screening tests reported.** | WBC, lymphocytes, monocytes, neutrophils, eosinophils, hemoglobin, platelets, sodium, potassium, urea, creatinine, AST, ALT, total protein, albumin | CLSI | Quintus 5-part Haematology analyzer (Boule Medical AB, Sweden); VITROS 350 analyzer (Ortho Clinical Diagnostics, USA) | No |

**Table 3.** (Continued)

| References | Years of study | Country/ies of study | N | Study population, recruitment, and type* | Ages** | Definition of "healthy"*** | Parameters measured | RI Methods **** | Analysis platforms | Comparisons across RI data sets / data sources |
|---|---|---|---|---|---|---|---|---|---|---|
| Smit 2019 [52] | 2012–2015 | South Africa | 711 | Adults (African, Caucasian and mixed race) recruited from the healthy South African population, no additional description of recruitment. | 18–65, 65+ to analyze variation | Participants were excluded if they suffered from any of the following conditions: (a) diabetes; (b) chronic liver or kidney disease; (c) had blood results which clearly indicated the presence of a severe disease; (d) had been hospitalized or been seriously ill during the previous 4 weeks; (e) donated blood in the previous 3 months; (f) were known carriers of hepatitis B virus (HBV), hepatitis C virus (HCV) or HIV; (g) were pregnant or within one year after childbirth; or (h) had participated in a clinical trial in the past 12 weeks. **No laboratory screening reported.** | Red cell count, Hemoglobin, Hematocrit, MCV, MCH, MCHC, RDW, White cells, Neutrophils, Lymphocytes, Monocytes, Eosinophils, Basophils, platelets | C-RIDL | Beckman Coulter ACT 5 diff AL analyzer | This study compared RIs within South Africa, between persons of Caucasian, Mixed Ancestry and African descent, ANOVA was used to test differences across ranges. The authors found that white cell, monocyte, neutrophil and red cell indices were significantly different statistically (p<0.05) amongst the three population groups, tending to be lower in Africans than values in persons of Caucasian or Mixed Ancestry (with the exception of some red cell indices, which were significantly different amongst the 3 groups). Comparisons were reported in a table and text. |
| Smit 2021 [53] | 2012–2015 | South Africa | 1,143 | 1,143 apparently healthy volunteers: 551 African and 592 non-African (comprising 383 Caucasian and 209 Mixed Ancestry). No details on sampling strategy reported. | 18–65 | Participants were excluded if they suffered from or had any of the following conditions: (i) diabetes under drug treatment; (ii) reported history of chronic liver or kidney disease; (iii) the presences of a severe disease as indicated by the results obtained in this study; (iv) had been hospitalised or seriously ill during four weeks preceding participation; (v) donated blood in the last three months; (vi) **a known carrier state of the hepatitis B virus (HBV), hepatitis C virus (HCV) or HIV-positive;** (vii) pregnant or within one year after giving birth and (viii) participated in another research study involving an investigational product during the past 12 weeks. | Total Protein, ALB, Urea, UA, Cr, Total Bilirubin, Na, K, Cl, Lactate to pyruvate, Glucose, Calcium, Phosphate, Magnesium, Cholesterol, HDL, LDL, Triglycerides, Thyroglobulin, Thyroglobulin antibodies, Thyroid peroxidase antibodies, Amylase, AST, ALT, LDH, ALP, GGT, CK, Lipase, Cholinesterase, CRP, Fe, Ferritin, TF, IgA, IgG, IgM, Free T4 (thyroxine), Free T3 (tri-iodothyronine), Thyroid-stimulating hormone, Prostate Specific Antigen | C-RIDL | automated Beckman Coulter DXC analyser for the samples obtained between 2012 and 2013 and using a Beckman Coulter AU analyser for the samples obtained in 2014. The analytes that depended on labelled immunoassays were measured by a Beckman Coulter Dxl analyser | No |
| Adoga 2012 [54] | Not reported | Nigeria | 1123 | Adults from urban and rural areas, no description of recruitment. | Mean age = 24.4, no ranges given | Participants who were **HIV, HBV or HCV** positive, or who had apparent signs or symptoms of ill-health were excluded from the study. Pregnant women and volunteers who indicated that they were on immunomodulatory drugs or had a history of idiopathic CD4 + T lymphocytopenia were also excluded. | CD4+, CD3+ T cells | Other | BD FACScount cytometer | No |

(Continued)

**Table 3.** (Continued)

| References | Years of study | Country/ies of study | N | Study population, recruitment, and type[*] | Ages[**] | Definition of "healthy"[***] | Parameters measured | RI Methods[****] | Analysis platforms | Comparisons across RI data sets / data sources |
|---|---|---|---|---|---|---|---|---|---|---|
| Troy 2012 [55] | Not reported | Zimbabwe | 542 | Infants as part of another study, no description of recruitment. Prospective study. | 3m, 5m, & 9m (same infants) | **HIV-infected** (PCR), reported as being currently ill on the questionnaire accompanying the blood draw, or with insufficient blood drawn to test for hematologic and immunologic values, were excluded. | WBC, lymphocytes, CD4+ T cells, hemoglobin, MCV, platelets | Other | Sysmex KX-21N Automated Hematology Analyzer (Sysmex Co. Ltd, Kobe, Japan); Partec Cyflow Counter (Partec GmbH, Munster, Germany) | For each value, the percentage of infants with values outside the normal range, as defined by a Text book published in the USA, was calculated. When the age of the study infants did not correspond to those in the reference normal ranges, ranges from adjacent ages were combined using the lowest and highest values to create the broadest range. The presence and degree of toxicities for WBC, hemoglobin and platelets were assessed using the 2009 update of the NIH Division of AIDS 2004 table for the severity of adult and pediatric adverse events. The presence and degree of immunodeficiency by CD4% was assessed using the World Health Organization (WHO) classification for immunodeficiency. Substantial proportions of the platelet counts (44%), hemoglobin results (19%) and mean corpuscular volumes (41%) were outside published normal ranges. The majority (71%) of CD4% values indicated immunodeficiency by World Health Organization criteria. Statistical testing was described but no results are presented. Comparison data are quantified in the text (but not presented in a table or figure). |
| Dosso 2012 [56] | Not reported | Ghana | 625 | Adults randomly selected from DSS. | 18–59 | General good health as determined by a clinician's medical history and physical examination, excluded pregnant women. **No laboratory screening tests reported** | Hemoglobin, Hematocrit, RBC, MCV, MCH, MCHC, RDW, Platelets, PDW, WBC, Lymphocytes, Monocytes, Granulocytes, ALT, AST, ALP, Amylase, Creatine Kinase, GGT, LDH, Total Protein, Albumin, Total Bilirubin, Direct Bilirubin, Cholesterol, Glucose, Iron, Triglycerides, Urea, Creatinine, Uric Acid, Chloride, Phosphorus, Potassium, Sodium | CLSI | Vitalab Selectra E Clinical Chemistry analyzer (Vital Scientific, The Netherlands); ABX Micros 60 analyzers (Horiba-ABX, Montpellier, France) | The authors compare their results to those from the platform package inserts (i.e., Caucasian), elsewhere in Ghana, Kenya, Tanzania, eastern and southern Africa (Kenya, Rwanda, Uganda and Zambia) and the USA. Compared to other references, the reference values for hemoglobin, hematocrit, red blood cell counts, and urea are lower in the Kintampo study area. Out of range values are shown compared to the package inserts. Using the biochemistry reference values based on the package inserts would have screened out up to 74% (e.g., creatine kinase) of potential male trial participants and >90% (e.g., monocytes) of the population using the hematology parameters. No statistical testing was reported. Comparisons were reported in a table and discussed in the text. |

*(Continued)*

**Table 3.** (Continued)

| References | Years of study | Country/ies of study | N | Study population, recruitment, and type* | Ages** | Definition of "healthy"*** | Parameters measured | RI Methods**** | Analysis platforms | Comparisons across RI data sets / data sources |
|---|---|---|---|---|---|---|---|---|---|---|
| Asare 2012 [57] | Not reported | Ghana | 4733 | Adults, randomly selected from Urban areas. | 25–65 | Those on medication, pregnant, or those who had diabetes, jaundice or renal failure just to mention a few (**complete list not reported**), were excluded. **No laboratory screening tests reported.** | Fasting Blood Glucose, 2HPP, total cholesterol, HDL cholesterol, LDL cholesterol, TG, UA, U, albumin, ALP | CLSI | Erba Smartlab Automatic Chemistry Analyzer (Erba, TransAsia, India) | Brief mention at the end of the discussion that the wide deviation from the manufacture's RIs supports our views that local RIs are critical to proper laboratory diagnosis. The assay platforms used in this study are from the UK and from India. No statistical testing was reported. Comparisons were reported in a table, differences were not quantified. |
| Chisale 2015 [58] | 2013 | Malawi | 105 | Adult blood donors, no description of recruitment. | 19–35 | Excluded those with detectable blood-borne infections such as **malaria, HIV, syphilis, or hepatitis**; females who were menstruating, pregnant, or lactating; those with any evidence of acute or chronic illness at the time of the study or in the preceding 3 months; those taking medications for any medical condition; those who smoked tobacco or consumed alcohol or had a current or past history of using illicit drugs; those who had experienced significant blood loss, from any cause (for example, surgery or trauma) within the preceding three months; those who had donated blood within the preceding three months; or those who had received a blood transfusion within the preceding 12 months | WBC counts, neutrophil, lymphocyte, monocyte, eosinophil, basophil, RBC counts (along with associated variables, such as Hb, Hct, MCV, MCH), platelets counts | Other | Beckman Coulter HmX haematology analyzer | The authors compared their data against reference limits from the machine used, which are based on a European population. The severity of "abnormalities" was classified using the Division of AIDS (DAIDS) criteria. Almost all variables generated some values that were outside the lower and upper reference limits provided by the manufacturers of the machine used. The proportions of out-of-range values ranged from 1.0% (1/105) to 88.6% (93/105). The majority of out-of-range results for Hb concentration and absolute counts of neutrophils, eosinophils and basophils were below the lower reference limits. In contrast, the majority of out-of-range results for platelet and monocyte counts were above the upper reference limits. Up to 55.0% (58/105) of our otherwise healthy study participants would have erroneously been considered to have at least one grade 1–4 hematological adverse event (AE). Some of the abnormal values for Hb concentration, neutrophil count, and platelets were classified as grade 3 (severe) or 4 (potentially life-threatening). No statistical testing was reported. |

(*Continued*)

**Table 3.** (Continued)

| References | Years of study | Country/ies of study | N | Study population, recruitment, and type* | Ages** | Definition of "healthy"*** | Parameters measured | RI Methods**** | Analysis platforms | Comparisons across RI data sets / data sources |
|---|---|---|---|---|---|---|---|---|---|---|
| Cumbane 2020 [59] | 2013–2014 | Mozambique | 419 | Adults at high risk for HIV enrolled in cohort study (RV363). | 18–35 | Healthy participants with evidence of **malaria, pregnancy, syphilis, hepatitis, and HIV** were excluded from this analysis. "**Healthy" was not otherwise defined** | Total bilirubin, direct bilirubin, ALT, creatinine, glucose, CD3+, CD8+, CD45+, CD4+ T cells, Hemoglobin, Hematocrit, Erythrocytes, Platelets, WBC, MCH, MCV, MCHC, Neutrophils, Lymphocytes, MXD, MPV, RDW_CV, RDW_SD | Other | Sysmex KX21N (Sysmex Corporation, Japan); Vitalab Selectra Junior1 (Vital Scientific, Netherlands); FACS Calibur flow cytometer (Becton Dickinson, USA) | Ranges were compared with those from Mozambique (persons at lower risk for HIV), South Africa, the USA, and the 2014 US National Institute of Health Division of AIDS (DAIDS) toxicity tables. The authors note relative differences across studies and quantify the number of out of range values when compared against the DAIDS tables. No statistical testing was reported. Comparisons were presented in a table and differences vs. DAIDS table were quantified. |
| Yalew 2016 [60] | 2014 | Ethiopia | 240 | Adult blood donors selected conveniently | 18–50 | The reference individuals were selected based on medical examinations, current health status, blood pressure, taking any medication, working with hazardous chemicals, alcohol intake, presence of inherited health disorder in the family, tuberculosis, lymphadenopathy, weight loss, regular exercise, tobacco smoking, allergy manifestation, fever, **malaria, HIV, hepatitis B virus surface antigen, hepatitis C virus antibodies,** menstruation, pregnancy and use of contraceptives. | | WBC, neutrophils, lymphocytes, eosinophils, monocytes, basophils, platelet count, RBC, Hb, hematocrit, MCV, MCH, MCHC, RDW | CLSI and other | Cell-Dyn 1800 | The authors compare their results to those from a textbook (Wintrobe's clinical hematology, derived from a Caucasian population), elsewhere in Ethiopia (Addis Ababa), Burkina Faso, Kenya, Togo, Tanzania, Jamaica, and Eastern and Southern Africa (Kenya, Uganda, Rwanda and Zambia). Median and 95th percentile of WBC for general population were lower than Caucasian population, Addis Ababa, Burkina Faso and Kenya of similar studies. The RBC, Hgb and PCV lower 95% limit values of both sex were lower than studies in Addis Ababa, Kenya, Burkina Faso and text book. While PCV upper limit values higher than the above countries. MCV values of the current study were higher than those countries while MCHC values were lower. Similarly, the absolute values of neutrophils in the current study were lower than Caucasian and Afro Caribbean but higher than African countries and Jamaica but lymphocyte count was higher. No statistical testing was reported. Comparisons were reported in a table and text, differences were not quantified. |

*(Continued)*

**Table 3.** (Continued)

| References | Years of study | Country/ies of study | N | Study population, recruitment, and type* | Ages** | Definition of "healthy"*** | Parameters measured | RI Methods**** | Analysis platforms | Comparisons across RI data sets / data sources |
|---|---|---|---|---|---|---|---|---|---|---|
| Serena 2019 [61] | 2014 | Mali | 161 | Adult male Malians as they arrived in Italy as immigrants (asylum seekers). | 19–54 | adult Malians, within 3 months of their arrival in Italy as asylum seekers, **HBsAg pos, anti-HCV pos, HIV pos** excluded. | RBC, Hb, Hct, MCV, MCH, MCHC, RDW, WBC, lymphocytes, neutrophils, monocytes, eosinophils, basophils, platelets, PDW, MPV, PCT, creatinine, glucose, AST, ALT | Other | Sysmex XN 1000 automated hematology analyzer; Dimension Vista 1500 Intelligent Lab System (Siemens) | The authors compared their results to those from the USA (including African Americans), Southern Europe/Italy, Kenya, Mozambique, Ghana, Tanzania, Botswana, Ethiopia, Central African Republic, Rwanda, Zambia, Uganda, Togo, South Africa, and Zimbabwe. The authors note many similarities, and report in the discussion on lower eosinophil and higher monocyte counts were observed in Malians compared to Europeans. No statistical testing was reported. Comparisons were reported in a table, differences were not quantified. |
| Ayemoba 2021 [62] | 2014 | Nigeria | 6,169 | 6169 of 7797 consenting male (96.2%) and female (3.8%) military applicants from 37 States of Nigeria. No description of sampling technique | 18–26 | Presence of glycosuria, proteinuria, haematuria and bilirubinuria—and **HBV, malaria, HIV,** pregnancy— screened out. No description of any screening questionnaires. | Assays performed included liver function tests (Total Serum Proteins, Albumin, Serum Globulin, alanine transaminase (ALT), aspartate transaminase (AST), Alkaline Phosphatase, gammaglutamyl transferase (GGT), Total Bilirubin, Direct and Indirect Bilirubin, Urea, Creatinine, Electrolytes (Na, K, Cl), Lipid profile (Total Cholesterols, HDL, LDL Cholesterols and Triglycerides), Calcium, Phosphate, Uric Acid, serum Lactate and serum Amylase) | CLSI | Selectra Pro-S1 automated clinical chemistry analyzer (Vital Scientific, Elitech Group Company, Netherlands). | Shown against western values (Dosso Ghana, American—Tietz Textbook of Clinical Chemistry and Molecular Diagnostics 2018, British—Clinical Biochemistry Reference Ranges Handbook 2017) however no quantification of differences made. No statistical testing was reported. |
| Ayemoba 2019 [63] | 2014–2017 | Nigeria | 6153 | Young adults screening for military service across the country. No description of recruitment. | 18–26 | apparently healthy (**not described**) testing for Hepatitis B surface antigen (**HBsAg), HIV-1 and 2, malaria** and pregnancy as exclusion criteria. | RBC, Hb, PCV, MCV, MCH, MCHC, Plt, WBC | Other | Sysmex KX-21-N hematology auto-analyzer (Sysmex Corporation Inc, Kobe, Japan). | The findings in this study and Western values obtained from Dacie and Lewis Practical Hematology text book are shown in Table 2, while Table 3 shows findings from this study and two neighboring West African countries [Ghana and Togo]. Median platelets count observed in this study (218 x 10⁹/l) was seen to markedly vary from Western Reference values (280 x 10⁹/l) (Table 2). Similarly, MCV was 84fl, 87fl and 85fl in this study and two other West African studies respectively, while Western value was 92fl (Tables 2 and 3). For the rest of the parameters, values from index study, neighboring Western African countries and Western values appeared similar. No statistical testing was reported. Comparisons were reported in a table, differences were not quantified. |

(*Continued*)

**Table 3.** (Continued)

| References | Years of study | Country/ies of study | N | Study population, recruitment, and type* | Ages** | Definition of "healthy"*** | Parameters measured | RI Methods**** | Analysis platforms | Comparisons across RI data sets / data sources |
|---|---|---|---|---|---|---|---|---|---|---|
| Osunkalu 2014 [64] | Not reported | Nigeria | 365 | Pregnant (n = 294) and not pregnant (n = 71) age-matched women attending the Nigerian Air Force Hospital, no description of recruitment. | Mean ages 28–30, no ranges given | Those with positive history and clinical evidence of past or present thrombo-embolic disorders, cardio-respiratory disorders, metabolic and chronic inflammatory conditions, history of bleeding disorders, use of contraceptives and non-steroidal anti-inflammatory drugs, smokers, and alcoholics were excluded from the study and also Injury requiring hospitalization or an emergency room management or special nonorthodox treatment within 4 weeks; surgery within the previous 4 weeks; current infection with fever >38°C; active menstruation; strenuous exercise within 12 hours were also excluded. All participants were those **screened negative for Human immune deficiency virus (HIV), Hepatitis B (HBsAg) and Hepatitis C** from hospital records. | D-dimer, PT, Platelet counts, INR (international normalized ratio based on PT results), aPTT, AST, ALT | Other | Hitachi chemistry analyzer | No |
| Miri-Dashe 2014 [65] | Not reported | Nigeria | 383 | Adults (blood donors) and pregnant women (ANC clinic attendees) selected during a blood drive. | 18–65 | Excluded positive for **HIV rapid test, HBsAg, HCV and Syphilis** or donated / received blood transfusion in the previous month. "Healthy" not well defined beyond lab tests. | WBC, RBC, Hb, PCV, platelets, lymphocytes, neutrophils, eosinophil, basophils, MCV, MCH, MCHC, Na, K, Cl, HCO3, Urea, Creatinine, Glucose, AST, ALT, Total Bilirubin, Amylase, Total cholesterol, Triglycerides | Other | Sysmex KX-21N (Sysmex Corporation Kobe, Japan); Vitros 350 fully automated Chemistry Analyzer (Ortho-clinical Diagnostic) | No |
| Genetu 2017 [66] | 2015 | Ethiopia | 200 | Pregnant women attending an ANC clinic, limited description of recruitment | 18–42 | **HIV negative, normal BMI (not defined)**, no other description of inclusion/ exclusion criteria | CD4+ T cells, Lymphocyte count, Total WBC count, Platelet count, RBC count, HB, Hematocrit, MCV, MCH, MCHC | Other | Cell-Dyn® 1800 (Abbott Laboratories, Chicago, IL, USA); BD FACSCount™ system (Becton Dickinson, San Jose, CA, USA). | Presented in the abstract and mentioned briefly in the discussion, the authors report that compared with the reference ranges derived from other studies, they found considerable variations in CD4 + T-cell count, HB, Hct, and MCV values. No statistical testing was reported. Comparisons were not presented visually and differences were not quantified. |
| Mulu 2017 [67] | 2015 | Ethiopia | 481 | Adults randomly selected for community based cross sectional survey | 18–65 | Age <18 years, adults with intestinal parasitic infections, hemo-parasites, skin rashes, history of blood transfusion < 6 months, **HIV positive, HBsAg positive, HCV positive,** pregnant, on any medication, exhibiting febrile symptoms, observable mental illness, disabled, smokers, chronic alcohol use, anemic, chronic diseases and acutely ill patients | WBC, RBC, Platelet, Lymphocyte, Neutrophil, Basophil, Eosinophil, Monocytes, HB, Hct, MCV, MCH, MCHC, CD4+ T cells | Other | Hematology analyzer MindrayBC320 (Mindray Biomedical electronic Corporation, China); FACS count (Becton Dickinson). | No |

(Continued)

**Table 3.** (Continued)

| References | Years of study | Country/ies of study | N | Study population, recruitment, and type* | Ages** | Definition of "healthy"*** | Parameters measured | RI Methods**** | Analysis platforms | Comparisons across RI data sets / data sources |
|---|---|---|---|---|---|---|---|---|---|---|
| Siraj 2018 [68] | 2015 | Eritrea | 591 | Adult college students, laboratory staff members, ACHS staff members, factory workers and school teachers | 18–49 | Physical examination and CLSI inclusion and exclusion criteria. The exclusion criteria included: the presence of any disease including; anemia, cardio-vascular disease and high blood pressure. Additional exclusion criteria included: drug therapy, alcohol consumption; heavy smokers, chronic diseases such as diabetic mellitus (DM); individuals who had donated blood in the last 3 months. Individuals who had undergone surgery in the recent past were also excluded Questionnaire provided, mentions pregnancy, menstruation, breast feeding (but no mention in paper of exclusion on these factors). Serological tests for hepatitis B virus (**HBV**) and hepatitis C virus (**HCV**) were undertaken. **HIV is not reported.** | RBC count, Hb, Hct, MCV, MCH, MCHC, RDW, WBC count, (lymphocytes, granulocytes, monocytes), platelet count | CLSI | AU 480 Chemistry System (Beckman Coulter) | The study RIs were compared to values currently in use by the Eritrean ministry of health (based on Beckman Coulter: AU 480 Chemistry System reagents inserts), and RIs from Tanzania, Ghana and elsewhere in Ethiopia. Overall, the proportion of OOR RIs ranged from 3.5 to 46.7%. In order of decreasing magnitude, the proportion of participants who were OOR included MCV (46.7%); MCH (37.9%) and Hb (36.2%). The lowest OOR proportions were observed in RBC (3.5%); platelet count (5.4%), Hct (11.1%), RDW (11.2%), total WBC count (12.5%) and MCHC (14.9%) respectively. This study had higher RBC RI compared to the study conducted in Ghana. However, the upper limit of the combined RBC RI in the Tanzanian study was comparatively higher. Further, the RI for Hgb (12.6–17.7 g/dL) and Hct (38.3–54.4%) were higher than those obtained from Tanzania and Ghana. No statistical testing was reported. Comparisons were reported in a table. |
| Alemu, 2019 [69] | 2015 | Ethiopia | 278 | Pregnant and age-matched, non pregnant women. The former were selected with systematic random sampling technique among ANC clinic attendees, the latter among ANC visitors and staff | Mean age 26, no range given | Apparently healthy; No previous history of diabetes mellitus, hypertension, and severe anemia, other chronic diseases, and taking medications that affect hematological indices for any reason were excluded. In addition, study participants who were smokers and chewers of chat were also excluded from the study. **No laboratory screening tests reported.** | WBC, Lymphocytes, Mixed cells, Neutrophils, Hb, Hct, RBC, MCV, MCH, MCHC, RDW, Platelet, PDW, MPV | Other | Sysmex KX-21N (Kobe, Japan) hematology analyzer | No |

*(Continued)*

**Table 3.** (Continued)

| References | Years of study | Country/ies of study | N | Study population, recruitment, and type* | Ages** | Definition of "healthy"*** | Parameters measured | RI Methods **** | Analysis platforms | Comparisons across RI data sets / data sources |
|---|---|---|---|---|---|---|---|---|---|---|
| Omuse 2018 [70] | 2015 | Kenya | 528 | Adults recruited from urban colleges, universities, churches, hospitals, corporations and shopping malls. (same population as Omuse 2020) | 18–65 | Exclusion criteria included individuals with a body mass index (BMI) greater than 35 kg/m2, consumption of ethanol greater than or equal to 70 g per day (equivalent to 5 alcoholic drinks), smoking more than 20 tobacco cigarettes per day, taking regular medication for a chronic disease (diabetes mellitus, hypertension, hyperlipidemia, allergic disorders, depression), recent (less than 15 days) recovery from acute illness, injury or surgery requiring hospitalization, known carrier state of **Hepatitis B, Hepatitis C or HIV**, pregnant or within 1 year after childbirth | RBC, Hb, Hct, MCV, MCH, MCHC, RDW, WBC, platelet, leukocyte differential counts of neutrophil, lymphocyte, monocyte, eosinophil, basophil | C-RIDL | Beckman Coulter ACT 5 DIFF CP analyser | The authors employed C-RIDL methods including larger sample size to facilitate comparisons with other studies. The present their data visually compared to 10 studies from eastern and southern Africa (Kenya, Rwanda, Uganda, and Zambia), Uganda (2 additional studies), Uganda, Ethiopia, Tanzania, Rwanda, Ghana, Togo, and South Africa. Their study highlights marked differences in certain CBC parameters such as Hb, eosinophil and platelet counts compared to other SSA countries. The authors also compare their neutrophil data to US RIs and note lower neutrophil counts compared to the US. No statistical testing was reported. Comparisons were reported in a Figure (neutrophil RIs are also specified in the discussion), differences were not quantified. |
| Omuse 2020 [71] | 2015 | Kenya | 533 | Adults recruited from urban colleges, universities, churches, hospitals, corporations and shopping malls. (same population as Omuse 2018) | 18–65 | Excluded: BMI >35 kg/m2, consumption of ethanol >70 g per day, smoking >20 cigarettes per day, taking regular medication for a chronic disease (diabetes mellitus, hypertension, hyperlipidemia, allergic disorders, depression), recent (< 15 days) recovery from acute illness, injury or surgery requiring hospitalization, carrier of **HBV, HCV or HIV**, pregnant or within 1 year after delivery | Sodium, Potassium, Chloride, Urea, Creatinine, Total Protein, Albumin, Total Bilirubin, Gamma-glutamyl transferase, Alkaline phosphatase, Lactate dehydrogenase, Calcium, Magnesium, Phosphate, Lipase, Total cholesterol, Triglycerides, High density lipoprotein cholesterol, Low density lipoprotein cholesterol, Uric acid, High sensitive c reactive protein, Amylase, Immunoglobulin A, Immunoglobulin G, Immunoglobulin M, Alanine aminotransferase, Aspartate aminotransferase, Creatinine kinase, Iron, Transferrin, Anti-thyroglobulin, Anti-thyroid peroxidase, Thyroid stimulating hormone, Free thyroxine, Free tri-iodothyronine, Ferritin, Prostate specific antigen | C-RIDL and other | Beckman Coulter AU 5800 (Brea, California, US); DXI (Brea, California, US) | Comparison of this study's RIs with those recommended by Beckman coulter and those derived from IFCC (i.e., similar methodology) studies conducted in India, Saudi Arabia and Turkey found much higher RIs for total bilirubin and c reactive protein. The authors also included in their supplementary material a table comparing their results against those from Kenya (three studies) and the USA. No statistical testing was reported. Comparisons were presented in a table; differences were not quantified. |

*(Continued)*

**Table 3.** (Continued)

| References | Years of study | Country/ies of study | N | Study population, recruitment, and type* | Ages** | Definition of "healthy"*** | Parameters measured | RI Methods**** | Analysis platforms | Comparisons across RI data sets / data sources |
|---|---|---|---|---|---|---|---|---|---|---|
| Fondoh 2020 [72] | 2015 | Cameroon | 340 | 340 of 487 voluntary blood donors who presented during the Regional Hospital Bamenda's voluntary blood donation programme. A stratified and clustered sampling method was used. The population was divided into two groups (men and women) and at least 50 samples were collected from participants at each of 3 sites and from each sex. 59% male | 18–65 (men), 18–60 (women) | The blood donors were subjected to several physical and medical screening protocols, as required by the national blood transfusion programme of the Ministry of Health, Cameroon, in addition to the CLSI recommendations. Criteria include: free from any non-communicable disease, should not have donated blood or had any STI in the previous three months, should not have been sick or been vaccinated during the previous four months and should not have been on any medication for at least a week before sample collection. Also, the donor should not have smoked on the day of donation or drunk alcohol for at least 24 hours before donation. Female donors should not be pregnant, breastfeeding, or on or expecting their menses within one week of the donation. Furthermore, the donor should have blood pressure of 100 mmHg– 140 mmHg/60 mmHg– 100 mmHg, weight greater than 50 kg, and temperature between 36.0˚C– 37.5˚C. **Exclusions for HIV, HBV, HCV or syphilis.** Participants who were sickle cell disease carriers (had the AS genotype) or who had sickle cell disease (had the SS genotype) were also excluded. | RBC; haemoglobin; haematocrit; mean cell volume; mean cell haemoglobin; mean cell haemoglobin concentration; coefficient of variation for the standard deviation of red cell distribution (%); standard deviation of red cell distribution; WBC; proportion of lymphocytes (%), monocytes (%) and granulocytes (%); absolute count of lymphocytes (×109/L), monocytes (×109/L) and granulocytes (×109/L); platelets; mean platelet volume; platelet distribution width and plateletcrit. | CLSI | Urit 3300 auto-analyser (Urit Medical Electronic [Group] Co., Ltd, Guilin, China). | The authors report that their findings are consistent with other studies in Africa: Addai-Mensah et al. in Ghana, Bakrim et al. in Morocco, Mulu et al. in Ethiopia, Miri-Dashe et al. in Nigeria, Dosoo et al. in Ghana and Kibaya et al. in Kenya. However, platelet counts in their study were relatively higher than those of other African countries in contrast to higher counts reported in the United States (as reported in Table 5, no quantification of differences made). |

(Continued)

**Table 3.** (Continued)

| References | Years of study | Country/ies of study | N | Study population, recruitment, and type* | Ages** | Definition of "healthy"*** | Parameters measured | RI Methods**** | Analysis platforms | Comparisons across RI data sets / data sources |
|---|---|---|---|---|---|---|---|---|---|---|
| Mekonnen 2017 [73] | 2015–2016 | Ethiopia | 446 | Adults randomly selected for community-based population | 18+, 3% > = 65 | intestinal parasitic infections, hemo-parasite, skin rashes, history of blood transfusion < 6 months, **HIV, HCV, HBV positives** and HCG positive (for females), observable mental illness, disabled, smokers, chronic alcohol use, anemic, malnourished (BMI<17.5Kg/m2), hospitalized persons, chronic diseases and acutely ill as per the recommendations of WHO | ALT, AST, ALP, amylase, LDH, creatinine, total protein, direct bilirubin, total bilirubin | Other | Mindray BS 200 clinical chemistry autoanalyzer (Germany) | The authors compare clinical chemistry parameters reference values between different African countries and USA (7 sets of RIs shown) with the current study (in the discussion section only, not in the results). The significant difference in gender (higher in male than females) in the reference values of ALT, AST and ALP in this study are consistent with reports from Botswana, Tanzania, Middle belt Ghana, Nigeria and Kenya. The reference values of ALT and AST are lower in the current study as compared to the reports from Botswana, Tanzania, Middle belt Ghana, Nigeria, Kenya. However, it is higher than those from USA. The reference interval of ALP is higher in this study compared to those from Tanzania, Middle belt Ghana, South Ghana and USA. The reference interval of amylase in this study is higher than those from Botswana, Tanzania, Middle belt Ghana, Nigeria, Kenya and USA. The reference interval of LDH in the current study is higher than those from Kenya, Tanzania, and USA, and lower than those from Middle belt Ghana. No statistical testing was reported. Comparisons were reported in a table, differences were not quantified. |

(*Continued*)

**Table 3.** (Continued)

| References | Years of study | Country/ies of study | N | Study population, recruitment, and type* | Ages** | Definition of "healthy"*** | Parameters measured | RI Methods**** | Analysis platforms | Comparisons across RI data sets / data sources |
|---|---|---|---|---|---|---|---|---|---|---|
| Oloume 2019 [74] | 2015–2016 | Cameroon | 294 | Adult blood donors via a stratified and clustered sampling strategy to enroll equal number of volunteers by blood bank & sex | 18–55 | Excluded: (1) currently sick or being followed up for a known pathology, (2) pregnant women, (3) undergoing treatment for any disease, (4) history of blood donation or transfusion within the immediate past 4 months, (5) regular smokers and (6) hospitalization within the immediate past month. In addition, we excluded people with a positive serology result for any one of the serological tests carried out at the blood bank, including **HIV, syphilis, hepatitis B virus, hepatitis C virus, positivity to blood parasites** through a thick or thin blood film, platelet aggregates, C-reactive protein > 6 mg/L, or inflammatory syndrome, as well as people with hyperthermia (> 37.8°C) or hypothermia (< 35°C) as determined by thermometer as a result of temperature control-related conditions | Hemoglobin, Red blood cells, Hematocrit, MCH, MCHC, MCV, Reticulocytes, Platelets, White blood cell, Neutrophils, Eosinophils, Lymphocytes, Monocytes, Basophils | Other | Pentra DX Nexus (Horiba-ABX, Montpellier, France) | No |
| Lawrie 2015 [75] | Not reported | South Africa | 381 | Children during a clinical pediatric study unpublished at the time, no description of recruitment. | 2w–12y | **HIV**-uninfected children, clinically healthy, no medications, and well-nourished as confirmed by doctor | WBC, lymphocytes, monocytes, eosinophils, neutrophils, basophils, RBC, Hb, Hct, MCV, RDW, platelets, absolute lymphocyte subsets using the following directly labelled antibodies CD3 APC, CD3 FITC, CD16 PE, CD19 FITC, CD45 PerCP, CD45RO PE, CD45RA FITC, HLA DR APC (BDIS), CD4 FITC, CD8 PE, CD56 PE | CLSI | FACSCalibur flow cytometer (Becton Dickinson, USA); Beckman Coulter LH-750 Haematology Analyzer (Beckman Coulter, USA) | The authors compare their results to those reference intervals currently in use by the South African NHLS. Although median and 95% CI values differed slightly, physiological trends for red cell, platelet, white blood cell differential and lymphocyte subsets were similar to international reference intervals currently in use at the NHLS, and support continued use of international pediatric reference intervals. No statistical testing was reported. Comparisons were discussed broadly in the text, differences were not quantified. |

(Continued)

**Table 3.** (Continued)

| References | Years of study | Country/ies of study | N | Study population, recruitment, and type* | Ages** | Definition of "healthy"*** | Parameters measured | RI Methods**** | Analysis platforms | Comparisons across RI data sets / data sources |
|---|---|---|---|---|---|---|---|---|---|---|
| Kieh 2020 [76] | 2015–2017 | Liberia | 3,223 | 2529 adults (44% female) and 694 children (53% female) recruited from two studies, a ph2 trial of an ebola vaccine and an ebola survivor & close contacts cohort study (PREVAIL I&III) | Three age groups (6–11, 12–17, and 18+ years) | The trial excluded 763 participants with a history of EVD (including survivors in PREVAIL III), those with a temperature of more than 38°C, and women who were pregnant or breastfeeding. The following additional exclusions were made for these analyses: (1) participants with a history of high blood pressure, diabetes or cancer; (2) participants with HIV or syphilis infection based on blood testing; and (3) participants with antibody levels against the Ebola virus surface glycoprotein considered indicative of past Ebola infection. Alcohol intake was not ascertained in either study. | a complete blood count was obtained with differential and platelet count, aspartate aminotransferase, alanine aminotransferase, creatinine, potassium, chloride and sodium | CLSI | Hematology using Cell Dyn Ruby (Abbott Diagnostics, Abbott Park, Illinois, United States), and chemistries using Trademark name for Alfa Wassermann's first chemistry analyzer (ACE) Alera or ACE Axcel (Alfa Wassermann, West Caldwell, New Jersey, United States) | The authors report that high percentage of both adults and children/adolescents would be classified as having high serum chloride levels based on the US reference standard (shown in Table 3 in the paper). Similarly, several haematology factors would be classified as out of range for both adults and children/adolescents using the US standard. A high percentage of both children/ adolescent and adult participants had haemoglobin, mean corpuscular volume, mean corpuscular haemoglobin, mean corpuscular haemoglobin concentration, and haematocrit levels below the lower limit. In addition, 20%–25% of Liberian adults had WBC counts and neutrophils below the US reference standard. No statistical testing was reported. Comparisons were reported in a table and text. |
| Yeshanew 2017 [77] | 2016 | Ethiopia | 400 | Pregnant women randomly sampled from ANC clinic attendees | 18–40 | Inclusion criteria included absence of active clinical disease conditions, being pregnant, and HIV-1/2 negative, syphilis negative and hepatitis B negative. | WBC, Neutrophil, Mixed, Lymphocyte, Platlet, CD4+, CD8+ T cells, RBC, Hb, Hct, MCV, MCH, MCHC | CLSI | CELL-DYN 1800; FACS Calibur system, Becton Dickinson Immunocytometry Systems | No |
| Achila 2017 [78] | 2016 | Eritrea | 255 | Adults conveniently sampled residing in capital | 60–80 | Male and female elders between 60 and 80 years old; healthy by doctor's examination; HCV and HBV sero-negative. excluded: diabetes, cardiovascular disease, neoplasm (cancer/malignancies, liver disease, kidney disease, medication, transfusion or recent surgery, heavy smoking, alcohol consumption, and significant recent illness. | AST, ALT, ALP, Cr, BUN, Total bilirubin, direct bilirubin, indirect bilirubin, total cholesterol, HDL, Na, Cl, K, CO2, albumin | CLSI | AU 480 Chemistry System (Beckman Coulter) | The RIs established in this study were compared to values currently in use by the Eritrean ministry of health, Ghana, Botswana, Eastern and Southern Africa (Kenya, Uganda, Rwanda and Zambia), and US Massachusetts General Hospital (MGH). Study RIs tended to be higher than RI from Caucasian populations, but lower than those obtained from reference interval studies in several African countries. Compared against Eritrean MoH data, as many as 31% (albumin) of some participants would have been called "out of range". CO2, HDL, and total cholesterol were all >20% "out of range". No statistical testing was reported. Comparisons were reported in a table and text. |

*(Continued)*

**Table 3.** (Continued)

| References | Years of study | Country/ies of study | N | Study population, recruitment, and type* | Ages** | Definition of "healthy"*** | Parameters measured | RI Methods **** | Analysis platforms | Comparisons across RI data sets / data sources |
|---|---|---|---|---|---|---|---|---|---|---|
| Onyekwelu 2021 [79] | 2016 | Nigeria | 240 | 240 of 245 women of reproductive age attending 6 primary health-care facilities in Abuja, Nigeria selected using multistage sampling technique (limited description of sampling technique) | 15–49 | Enrolled women without pre-existing medical conditions such as chronic hypertension, not on any form of calcium supplementation and had no bone metastasis of carcinomas. Women on tamoxifen, thiazide and those who were pregnant were also excluded. Five participants were eventually excluded from the analysis due to obesity and underweight. **No laboratory screening tests reported.** | Serum total protein, Serum albumin, Measured Total calcium, Corrected total calcium | CLSI | The serum PH, ionised calcium and total calcium were measured using ionised selective electrodes (Stat-profile PrimeR-Novabiomedical), while serum total calcium, protein and albumin were assayed spectrophotometrically (SpectronicR 20D). | No |
| Abebe 2018 [80] | 2016–2017 | Ethiopia | 1175 | Adult blood donors | 18–60 | Examination, disease history, medications review to exclude chronic alcohol abusers, smokers, pregnant and lactating women, positive for **HIV, HBV, HCV, and syphilis,** and who had a history of jaundice within 3 months and major surgery within 1 year. | Alanine aminotransferase, aspartate aminotransferase, ALP, GGT, amylase, creatinine, urea, uric acid, total cholesterol, triglycerides, total protein, albumin, direct bilirubin, total bilirubin | CLSI | Mindray BS-200E full automated clinical chemistry analyzer | No |
| de Koker 2021 [81] | 2016–2017 | South Africa | 662 | 662 of 774 prospective voluntary blood donors sampled conveniently, 62% female | 18–60 | Completed blood donor screening & questionnaire, had to meet routine donor criteria, including irregular blood pressure or pulse rate. **Screened and excluded for HBV, HCV, HIV, syphilis.** No mention of pregnancy (presumed not pregnant) | Hb, RCC, Hct, MCV, mean cell Hb, MCHC, platelet count, WBC and 5-part differential (neutrophils, lymphocytes, monocytes, eosinophils, basophils), serum ferratin | CLSI | XN-1000 instrument (Sysnex Corp, Japan) | Compared with data from two studies and the NHLS, all from South Africa. No statistical testing was reported. No quantification of differences is made. |
| Eshete 2016 [82] | Not reported | Ethiopia | 336 | Adult blood donors selected using convenient sampling methods | 18–58 | No description of health, **no laboratory screening tests reported.** 5.2% were excluded "due to infection." (no additional details provided) | White blood cell, Red blood cell, Hemoglobin, Hematocrit, Mean corpuscular volume, Mean corpuscular hemoglobin, Mean corpuscular hemoglobin concentration, Mean platelet volume, Platelet count, Total cholesterol, Triglyceride, HDL, LDL | Other | Automated Sysmex KX 21 hematology analyzer; Cobas Integra 400 plus clinical chemistry analyzer. | Compared against reference values being used in clinical practice (no details provided—"values being used in clinical practice"), including comparisons by sex, age. Multiple tests reported (t- test for internal comparisons across gender, Mann-Whitney, Kruskal-Wallis rank test, and kennel's correlation). The authors report that hematological and lipid variables obtained were statistically significantly different from the reference range currently used in clinical practice, however it was unclear if the statistical testing done was valid. Data are presented in a table. |

(*Continued*)

**Table 3.** (Continued)

| References | Years of study | Country/ies of study | N | Study population, recruitment, and type* | Ages** | Definition of "healthy"*** | Parameters measured | RI Methods**** | Analysis platforms | Comparisons across RI data sets / data sources |
|---|---|---|---|---|---|---|---|---|---|---|
| Bimerew 2018 [83] | 2017 | Ethiopia | 883 | Children, adults, geriatric recruited via non-probability convenience sampling technique from schools, universities, employees, pensioners, and other volunteers residing in three towns | 5–71, stratified by age, but strata not defined | Apparently healthy individuals aged 5 years and above who have lived in the study areas for more than 1 year were included in the study. We excluded individuals who had a positive result from the screening tests (C-reactive protein (**CRP**), hepatitis B surface antigen (**HBsAg**), anti-hepatitis C virus (**HCV**) antibody). In addition, individuals with known acute and chronic diseases; known history of hematologic disorders; recent history of blood loss; blood donation in the last 6 months; blood transfusion in the previous year; immunization in the last 6 months; major surgical procedures in the past 6 months; those taking pharmacologically active agents including oral contraceptives; replacement or supplementation therapy such as insulin; smokers; and pregnant women were also excluded. | WBC, RBC, Hb, Hct, MCV, MCH, MCHC, PLT, RDW-CV, Neutrophil, Lymphocyte, Monocyte, Eosinophil, Basophil | CLSI | Sysmex XS-500i hematology analyzer (Sysmex Corporation Kobe, Japan) | No |
| Bawua 2020 [84] | 2017 | Ghana | 501 | Adults recruited from public institutions, schools, health facilities, social centers, churches, and mosques | 18–70 | The selection of eligible participants was based on well-defined inclusion and exclusion criteria, in accordance with the IFCC/C-RIDL protocol. The exclusion criteria included the following: individuals with known diabetes mellitus under drug therapy, recent (≤14 days) recovery from acute illness, injury, or surgery requiring hospitalization, known carrier state of **hepatitis B virus, hepatitis C virus, or HIV**, pregnancy or within one year after childbirth. However, considering a Ghanaian general standard of healthiness, additional exclusion criteria not stipulated in the C-RIDL protocol were set to exclude individuals with BMI ≥ 35 kg/m2, consumption of alcohol ≥70 g/d, smoking habit of >20 cigarettes/d. | RBC, Hb, Hct, MCV, MCHC, RDW, platelet, MPV, WBC, neutrophils, eosinophils, lymphocytes, basophils, monocytes, IgG, IgA, IgM, C3, folate, TF, vitamin B12, Fe, ferritin, CRP | C-RIDL | Sysmex XN-1000 analyzer; Beckman-Coulter/AU480; Centaur-XP/Siemens | The authors compared their RIs to those from Kenya, Uganda, Zimbabwe, Morocco, Turkey, Malaysia, China, Australia, Spain, Japan, and the USA. Global comparison of Ghanaian RIs revealed lower-side shift of RIs for leukocyte and neutrophil counts, female hemoglobin, and male platelet count, especially compared to non-African countries. No statistical testing is reported to compare RIs across regions/ studies. Data are presented in figures (except data from Japan), but not quantified (i.e., the numbers and percentages out of range are not reported). |

*(Continued)*

**Table 3.** (Continued)

| References | Years of study | Country/ies of study | N | Study population, recruitment, and type* | Ages** | Definition of "healthy"*** | Parameters measured | RI Methods**** | Analysis platforms | Comparisons across RI data sets / data sources |
|---|---|---|---|---|---|---|---|---|---|---|
| Sing'oei 2021 [85] | 2017–2018 | Kenya | 299 | 299 of 579 (many excluded for missing data) from a prospective observational HIV incidence cohort study. Limited details on the sampling strategy were reported (from the discussion: self-selected population, representative of participants willing to participate in a clinical trial, enrolling adults without HIV from the general population including fisher folk communities and sex workers) | 18–35 | All participants underwent medical history-taking, physical examination and tests for **HIV, malaria, syphilis, schistosomiasis, and hepatitis B and C**. Female participants also underwent pregnancy testing. Participants who were living with HIV, pregnant, or had significant medical conditions were excluded from the study. Additionally, participants with syphilis, hepatitis B, hepatitis C, or malaria, and/or who were missing key laboratory result were excluded from the analyses to establish reference laboratory values for a healthy population. | Absolute WBC counts and percentages for WBC differentials (neutrophils, lymphocytes, monocytes, eosinophils, and basophils), RBC with related parameters (hemoglobin, hematocrit, MCV, MCH, MCHC, and RDW), and platelet counts; clinical chemistries included ALT and creatinine; CD4 T cell counts were also done. | CLSI | Hematology parameters were determined from whole blood using a Coulter Ac•T™ 5diff CP analyzer (Beckman Coulter, Paris, France). Chemistries were determined using the cobas[R]c 311 biochemistry analyzer (Roche, Mannheim, Germany). CD4 counts were determined using the point of care Pima™ Analyzer (Alere Technologies, Jena, Germany) | The authors comare their results to three other studies in Kenya, and results from the USA. No statistical testing was reported. Comparisons were reported in a table, selected differences were quantified in the text. |
| Addai-Mensah 2019 [86] | 2018 | Ghana | 488 | Adult blood donors, no description of recruitment | 18–60 | Females were not pregnant. Individuals with confounders like malaria, glucose-6-phospate dehydrogenase deficiency, and sickle cell disease were excluded. **No laboratory screening tests reported.** | WBC, RBC, Hemoglobin, Hct, MCV, MCH, MCHC, RDW-CV, Platelet, Neutrophils, Lymphocytes, Monocytes, Eosinophils, Basophils | Other | Sysmex KX 4000i hematology analyzer (Sysmex Corporation, Kobe, Japan) | The authors observed in our study in the discussion that the hematological reference ranges (Tables 2 and 3) were lower compared to the accompanying manual of the hematology analyzer used in the study. No statistical testing was reported. Comparisons were not presented visually and differences were not quantified. |
| Belay 2020 [87] | 2018–2019 | Ethiopia | 344 | Adolescents in community based cross sectional pop from randomly selected urban areas | 12–17 | excluded: known chronic illnesses like diabetes mellitus, chronic renal insufficiency, hypertension, ischemic heart disease, anemia, thyroid, liver diseases, and cancer, **HIV, Hepatitis,** taking pharmacologically active substances and all prescription drugs, who had **Hemo-parasite and intestinal parasite,** who received blood transfusion within the previous 1 year, Pregnant females | FBS, ALT, AST, ALP, Creatinine, urea, total protein, albumin, direct bilirubin, indirect bilirubin | CLSI | Biosystem 25 A (Biosystem, Spain) clinical chemistry analyzer | No |
| Haileslasie 2020 [88] | 2018–2019 | Ethiopia | 249 | Adolescents, community based, using well described random selection | 12-17y | Individuals with any chronic and acute illnesses, taking antibiotic treatment, recent history of blood loss, blood transfusion in the last one year, immunization in the last 6 months, major surgical procedures in the past 6 months, who have any intestinal and hemo-parasites were excluded during data analysis. **No laboratory screening tests reported** | RBC, WBC, Hb, Hct, MCV, MCH, MCHC, Platelet, RDW-CV, Neutrophil, Lymphocyte, MPV | Other | Sysmex KX-21N hematology analyzer (Sysmex Corporation Kobe, Japan) | No |

*(Continued)*

**Table 3.** (Continued)

| References | Years of study | Country/ies of study | N | Study population, recruitment, and type* | Ages** | Definition of "healthy"*** | Parameters measured | RI Methods**** | Analysis platforms | Comparisons across RI data sets / data sources |
|---|---|---|---|---|---|---|---|---|---|---|
| Enawgaw 2018 [89] | Not reported | Ethiopia | 967 | Adult blood donors selected by convenience sampling | 18–61 | Good health assessed through exam and questionnaire. Donors were screened for infectious diseases (**HIV, Hepatitis B virus, Hepatitis C virus and syphilis**) by blood bank. We excluded individuals known to have diabetes mellitus, chronic renal insufficiency, hypertension, ischemic heart disease, anemia, thyroid or liver disease; those taking pharmacologically active substances or any prescription drugs; smokers; individuals who had malaria in the previous 3 months, individuals who had jaundice or major surgery in the past year; pregnant (determined clinically or by urine HCG test) and lactating women; individuals who had donated blood in the previous 4 months and those who had received a blood transfusion in the previous year. | WBC, platelet, RBC, Hb, PCV, CD4+ T cells, MCV, MCH, MCHC, RDW, MPV | Other | CELL-DYN 1800 (Abbott Laboratories Diagnostic Division, USA); BD FACS Count system (Becton Dickinson, San Jose, CA, USA) | Presented in the discussion, the authors compare their results to RIs from elsewhere in Ethiopia, Kenya, eastern and southern Africa, Ghana, Tanzania, Togo the USA and the textbook Rodak's Hematology (published in the USA). No statistical testing was reported. Comparisons were reported in a table, differences were not quantified. |
| Gessese 2020 [90] | 2019 | Ethiopia | 329 | Adult blood donors, no description of recruitment | 18–65 | Apparently healthy without a positive result from the screening tests (**hepatitis B surface antigen, anti-hepatitis C virus antibody**). In addition, individuals with known acute and chronic diseases, known history of hematologic disorders, recent history of blood loss, blood donation in the last 6 months, blood transfusion in the last 6 months, immunization in the previous year, major surgical procedures in the past 6 months, those taking pharmacologically active agents including oral contraceptives, replacement or supplementation therapy such as insulin, smokers, and pregnant women were al-so excluded. **HIV is not reported.** | Red blood cell, hemoglobin, hematocrit, mean corpuscular volume, mean corpuscular hemoglobin, mean corpuscular hemoglobin concentration, red cell distribution, white blood cells, lymphocytes, mixed, neutrophils, platelets | CLSI and other | Sysmex XP-300 | The RIs in this study were different from previous studies which were conducted in different regions of Ethiopia or African countries (Kenya, Ghana, Uganda, Eastern and Southern Africa) or in the USA, and a text book (Rodak's Hematology). No statistical testing was reported. Comparisons were reported in a table, differences were not quantified. |

(*Continued*)

**Table 3.** (Continued)

| References | Years of study | Country/ies of study | N | Study population, recruitment, and type* | Ages** | Definition of "healthy"*** | Parameters measured | RI Methods**** | Analysis platforms | Comparisons across RI data sets / data sources |
|---|---|---|---|---|---|---|---|---|---|---|
| Tiruneh 2020 [91] | 2019 | Ethiopia | 151 | Newborns selected by simple random sample attending OBGYN clinic at university hospital | newborn | The newborns were selected based on the following criteria: Birth at full-term (39–42 weeks of gestation), and absence of any congenital anomalies. All selected newborns were physically examined at birth and found normal and apparently healthy. The premature newborn (delivered less than 37 weeks of gestation), twin newborns, the pregnancy complicated with diabetics, preeclampsia, hypertension, HIV/ AIDS, chronic kidney, liver disease, malaria, anemia, and hematological malignancy were excluded from the study. On the other hand, a mother who had bleeding during pregnancy, maternal drinking of alcohol during pregnancy, cigarette smoking during pregnancy, and no antenatal care also were excluded from being sampled. **Presumably mothers were tested for HIV, no laboratory screening tests reported for the infant.** | WBC, neutrophils, lymphocytes, eosinophils, monocytes, basophils, platelet count, RBC, Hb, hematocrit, MCV, MCH, MCHC, RDW | CLSI | Sysmex KX-21N (Sysmex Corporation Kobe, Japan) automated analyzer | The authors compare their RIs to results published from Nigeria (2 studies), Iraq, Pakistan, Nepal, Saudi Arabia, and Iran. The Hgb values of this study almost agrees with a study reported in Saudi Arabia. The RI of RBC of the current study was comparable to Nepal, Iraq and Nigeria findings. However, the lower limit of WBC was lower than a study reported in Saudi Arabia, Nepal, and Pakistan. No statistical testing was reported. Comparisons were reported in a table, differences were not quantified. |
| Sissay 2021 [92] | 2019 | Ethiopia | 472 | 472 of 513 apparently healthy adults of both sexes and pregnant women (n = 157). Volunteers were recruited by outreach workers in randomly selected communities to get sample representative of the region. | non-pregnant: 18–65, pregnant: 15–49 | Subjectively feel healthy, and have normal records of vital sign and physical examination. All participants were screened for **malaria, HIV, Hepatitis B surface antigen, Hepatitis C antibody, Treponema pallidum.** CRP was measured using a qualitative test, abnormal CRP results screened out. Also excluded: hypertension, diabetes mellitus, chronic gastritis, cancer, cardiac illness, bleeding disorders, allergy, anemia, and any unspecified chronic diseases history of blood transfusion within the last one year or donated blood within the last six months, hospital admission for the last one year, had surgery in the last three years, had malaria in the last six months, and diagnosed with any form of TB in the last two years; Individuals who were drug abusers, had work exposure to hazardous chemicals, smokers, and more than occasional (holidays, special ceremonies) alcohol drinkers and Khat (Catha edulis) chewers. Pregnant women with active bleeding during the data collection period encountered pregnancy and obstetrics complication; and non-pregnant women who were menstruating during the data collection period; whose average menstruation stay >7days; or have taken over the counter of oral contraceptive and are breast feeding were also excluded from the study. | WBC, mixed cells count, granulocytes, lymphocytes, RBC, Hgb, Hct, MCV, MCH, MCHC, RDW, platelets count, MPV, and PDW. | CLSI | Mindray BC-3000 Plus automated hematological analyzer (Mindray Bio-medical electronics Co., Ltd., Shenzhen, China) | The authors comare their results to 'current practice' RIs in use regionally, included in the package insert of the analyzer (additional details on source population(s) not reported). Percent out of range values from the current study are shown against current practice RIs. These results are also shown against results from Ghana, Zimbabwe, Mali, and Eritrea, though out of range percentages are not shown for these comparisons. No statistical testing was reported. Comparisons were reported in a table, with selected differences were quantified in the text. |

*(Continued)*

**Table 3.** (Continued)

| References | Years of study | Country/ies of study | N | Study population, recruitment, and type* | Ages** | Definition of "healthy"*** | Parameters measured | RI Methods **** | Analysis platforms | Comparisons across RI data sets / data sources |
|---|---|---|---|---|---|---|---|---|---|---|
| Gebereh 2021 [93] | 2019 | Ethiopia | 360 | 360 of 600 healthy pregnant women recruited by convenience sampling technique | 18–45 | Well-defined exclusion and partitioning criteria (gestational age, in trimesters, 120 women in each trimester) before the selection of the reference individual. Exclusions included current health status, blood pressure, diabetics, treatment (medication), working with hazardous chemicals, alcohol intake, presence of inherited health disorder in the family, tuberculosis, lymphadenopathy, weight loss, regular exercise, tobacco smoking, allergy manifestation, fever and infectious diseases (i.e., **malaria, HIV, hepatitis B virus, hepatitis C virus, syphilis**) and history of chronic illness. | Differential count (Neutrophils, Lymphocyte, Eosinophil, Monocyte, and Basophil), Hb, Hct, MCV, MCH, MCHC. The immunological parameters were CD4+ cell absolute counts, CD4+ cell percentage | CLSI | Cell-Dyn Ruby (Abbott Laboratories, Chicago, IL, USA) a five parameter automated hematology analyzer; BD FACS presto cartridge (Becton Dickinson, San Jose, CA, USA) | No |
| Angelo 2021 [94] | 2019 | Ethiopia | 139 | healthy, term newborns with normal birth weight born to apparently healthy pregnant mothers who had met the eligibility criteria. Convenient non-probability sampling technique | newborn (37–42 weeks) | mothers aged 18 to 45 years were recruited in the study. Mothers with the following conditions were excluded: those with medical conditions like infectious (e.g. **Hepatitis B, HIV, Syphilis**), chronic illness (e.g. Insulin-dependent diabetes mellitus), obstetric (e.g. less than six months from abortion, preeclampsia), psychological problems and social habits (e.g. smoking, heavily alcohol drinking). | WBC, RBC, HGB, HCT, MCV, MCH, MCHC, PLT, LYM, MXD, NEU, RDW, PDW, and MPV | CLSI | Sysmex KX-21 N (Sysmex Corporation, Kobe, Japan) | Compared against the RIs included with Sysmex KX 21, RIs from Sudan, Nigeria, Saudi Arabia, Pakistan. No statistical testing was reported. No quantification of differences is made. |
| Abbam 2021 [95] | 2019–2020 | Ghana | 992 | population-based cross-sectional study, randomly selected healthy voluntary blood donors from the four eco-geographic zones (Coastal Savannah, Rain Forest, Savannah and Transitional) | 18–59 | general health status questionnaire (adapted from the CLSI Guidance Document C28A2). Pregnant and breastfeeding mothers, obesity (BMI >29 kg/m2), evidence of medication use, use or abuse of alcohol and tobacco, presence of acute/chronic disease conditions, history of blood donation or transfusion within the last 3 months, surgery or hospitalization within the last 1 to 6 months, incomplete laboratory analysis results and any other confounding factors that may compromise the assessment of the analytes of interest were excluded from the study. G6PD deficiency, asymptomatic or sub-clinical **malaria, Hepatitis B and C, HIV**, sickle cell and other abnormal haemoglobin variants were excluded | RBC, Hb, HCT, MCV, MCH, MCHC, RDW-CV and RDW-SD, platelet count, WBC, lymphocyte counts-absolute and percentage, monocyte count- absolute and percentage, neutrophil counts-absolute and percentage, eosinophil counts-absolute and percentage, and basophil counts-absolute and percentage. AST, ALT, ALP, GGT, BID, direct bilirubin, albumin, total protein, total cholesterol, triglyceride, high density lipoprotein cholesterol, low density lipoprotein cholesterol, urea, and creatinine levels | CLSI | YUMIZEN H500 (5-part differential) haematology auto analyzer (HORIBA ABX, France), DIALAB Autolyser (DIALAB GmbH, Austria) | Compared by eco-geographical zones within Ghana. The percentage out of range (OOR) values of the study RIs were computed, the proportion of normal Ghanaian adults whose haematology and biochemistry laboratory results would have been described as abnormal when the accompanying RIs provided by HORIBA (haematological) and DIALAB (biochemistry) are used. No statistical testing was reported. |

*(Continued)*

**Table 3.** (Continued)

| References | Years of study | Country/ies of study | N | Study population, recruitment, and type* | Ages** | Definition of "healthy"*** | Parameters measured | RI Methods **** | Analysis platforms | Comparisons across RI data sets / data sources |
|---|---|---|---|---|---|---|---|---|---|---|
| Nlinwe 2021 [96] | 2020 | Cameroon | 350 | The study participants were consecutively recruited from apparently healthy voluntary nonremunerated blood donors at the Bamenda Regional Hospital | 18–60 | All pregnant/ lactating/ menstruating women, those who were underweight (body mass index of ≤18.5), had lost >10% body weight in the past 6 months, had jaundice in the past 12 months, had a blood transfusion, sickle cell anemia, on any medication, and/or had an unexplained fever in the past 3 months were excluded from the study. Also, all those who tested positive for **hepatitis B virus (HBV), hepatitis C virus (HCV), malaria (Plasmodium falciparum antigen histidine-rich protein 2 rapid diagnostic test), Treponema pallidum hemagglutination (syphilis), and HIV 1/2 antibodies (HIV 1/2 antibodies)** were excluded from the study. | RBC, WBC, differential count (lymphocyte, monocytes, granulocytes [combined tally of neutrophils, eosinophils, and basophils]) HGB, HCT, MCV, MCH, MCHC, RDW_CV, RDW_SD, PLT, MPV, PDW, PCT | CLSI | Hematology done with the Urit 3300 autoanalyser (Urit Medical Electronic (Group) Co., Ltd, Guilin, China) | No |
| Payne 2020 [97] | Not reported | South Africa | 381 | Children from a Child Wellness Clinic in informal settlement, attendance was voluntary, limited description of recruitment | 2w–13y; multiple different stratifications to enable comparison to other published data sets | Attendance at the CWC was voluntary, and the criteria for recruitment were that the child was well at the time with no chronic medical condition or prescription medications, registered at the health clinic, and attended with their biological mother and hand-held medical record. Maternal HIV-exposure was not excluded as. **HIV positive** children were not included. | Specific lymphoid subsets assessed included total CD3+, CD3+/CD4+, CD3+/CD8+, CD3-/CD56+, CD16+/56+, CD3+/HLA DR+, CD3 +/CD4+/HLA DR+, CD3 +/CD8+/HLA DR+, CD3 +/CD4+/45RA+, CD3+/CD4 +/45RO+, CD3+/CD8 +/45RA+, CD3+/CD8 +/45RO+, CD19+ | Other | Becton Dickinson FACSCalibur | The goal of this work was to compare immunological reference intervals for children from Europe (2 studies, Germany and The Netherlands) and the US (1 study) with South African children to explore whether healthy children living in settings with high rates of infectious diseases have different baseline immunological parameters. Increased activated T-cells, and natural killer cells were seen in the younger age-groups. The main finding across all age-groups was that the ratio of naïve/memory CD4 and CD8 T-cells reached a 1:1 ratio around the first decade of life in healthy South African children, far earlier than in resource-rich countries, where it occurs around the fourth decade of life. The authors present their data according to the presentation of data in each publication. The data are presented visually (tables and figures), and where possible statistical tools are employed for comparisons. |

Data are sorted by year of study, or year of publication where year of study was not reported (n = 13).

* All studies cross sectional except where noted.

** Ages shown by stratification, where provided.

*** Including laboratory screening for infectious diseases (shown in bold).

**** Where specified, either CLSI, C-RIDL or 'other'.

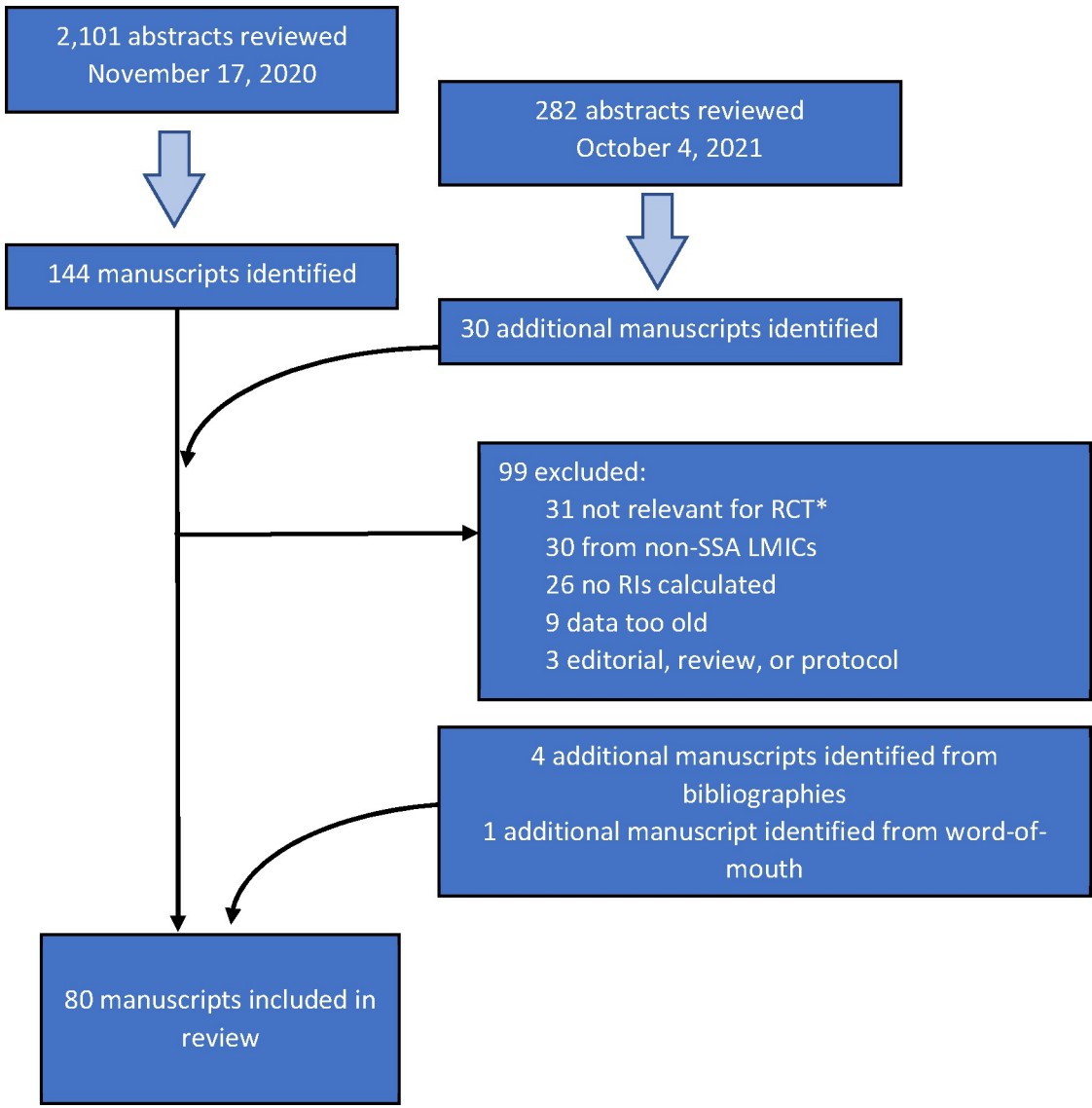

**Fig 1. Flowchart of manuscripts identified and reasons for exclusion.** * Included studies of how values varied under differing laboratory/storage conditions, studies relevant for comparative patient care but not RCTs, studies of lung function, focus on decision limits and not RIs etc. LMIC: Lower Middle-Income Country; RIs: Reference Intervals; RCT: Randomized Controlled Trial; SSA: sub-Saharan Africa.

Thirteen studies (16%) omitted the year(s) when the study was conducted. In two cases, hematology and chemistry were published separately for the same study population (i.e., four publications, two study populations). We found no manuscripts including RIs for people living with HIV.

Recruitment, screening, and enrollment procedures varied significantly across studies. Blood donors represented the largest source of volunteers, with 15 (19%) studies recruiting from persons donating blood. Fourteen studies (18%) reported recruiting volunteers for RIs from screening and/or enrollment into clinical trials or observational studies. Only 23 (29%) studies reported efforts to enroll a representative sample, describing methods to randomly select volunteers from a well-defined source population. All studies attempted to enroll

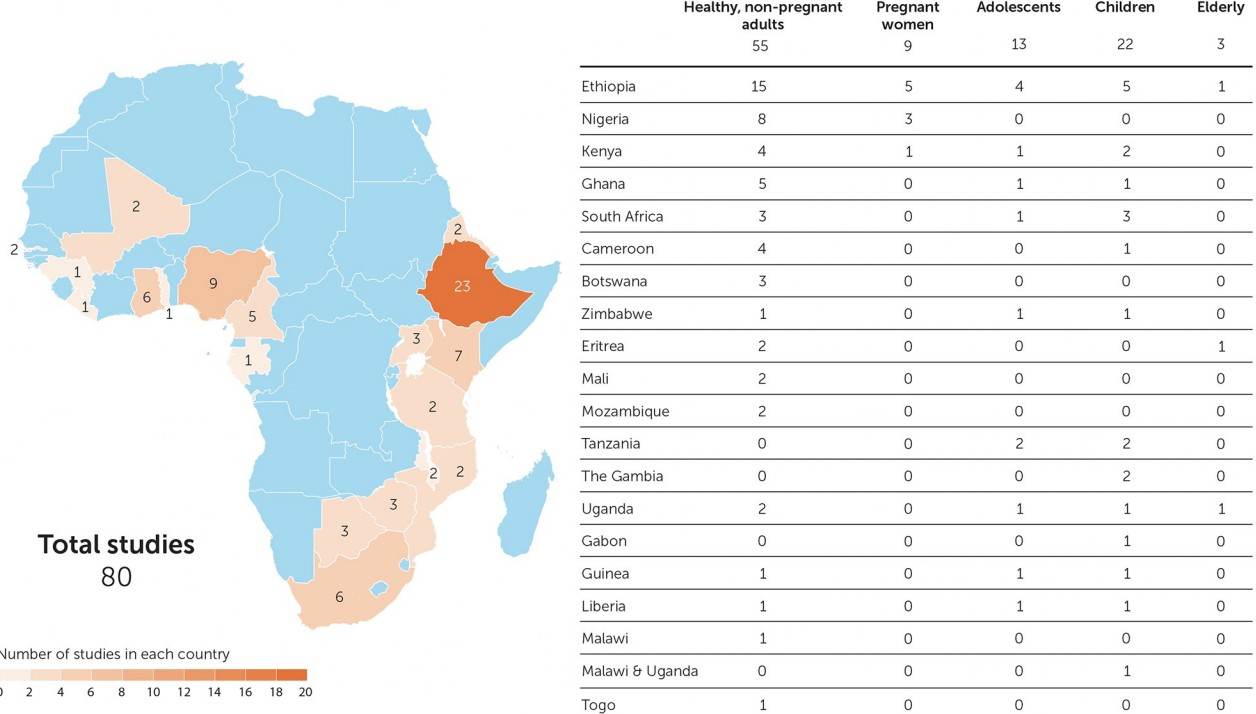

| | Healthy, non-pregnant adults | Pregnant women | Adolescents | Children | Elderly |
|---|---|---|---|---|---|
| | 55 | 9 | 13 | 22 | 3 |
| Ethiopia | 15 | 5 | 4 | 5 | 1 |
| Nigeria | 8 | 3 | 0 | 0 | 0 |
| Kenya | 4 | 1 | 1 | 2 | 0 |
| Ghana | 5 | 0 | 1 | 1 | 0 |
| South Africa | 3 | 0 | 1 | 3 | 0 |
| Cameroon | 4 | 0 | 0 | 1 | 0 |
| Botswana | 3 | 0 | 0 | 0 | 0 |
| Zimbabwe | 1 | 0 | 1 | 1 | 0 |
| Eritrea | 2 | 0 | 0 | 0 | 1 |
| Mali | 2 | 0 | 0 | 0 | 0 |
| Mozambique | 2 | 0 | 0 | 0 | 0 |
| Tanzania | 0 | 0 | 2 | 2 | 0 |
| The Gambia | 0 | 0 | 0 | 2 | 0 |
| Uganda | 2 | 0 | 1 | 1 | 1 |
| Gabon | 0 | 0 | 0 | 1 | 0 |
| Guinea | 1 | 0 | 1 | 1 | 0 |
| Liberia | 1 | 0 | 1 | 1 | 0 |
| Malawi | 1 | 0 | 0 | 0 | 0 |
| Malawi & Uganda | 0 | 0 | 0 | 1 | 0 |
| Togo | 1 | 0 | 0 | 0 | 0 |

**Fig 2. Geography and study populations included in 80 African studies of reference intervals.** Only one study included data from more than one country, Malawi, and Uganda. Rows and columns do not sum to total studies (80) as some studies include multiple study populations.

"healthy" volunteers, however the description of inclusion and exclusion criteria varied widely. Those studies that drew from volunteers of other studies or from blood donors may have involved prescreening on "health", however this was often not reported (Table 3). Laboratory tests employed in screening volunteers for participation varied and was sometimes not made explicit. As reported in the reviewed papers, 53 (66%) studies screened for HIV (including maternal screening, where infants were enrolled), 37 (46%) for hepatitis B, 29 (36%) for hepatitis C, 20 (25%) for malaria, 18 (23%) for sexually transmitted infections (STI- typically syphilis), 4 (5%) for intestinal parasites (often not specified), and 1 (1%) for hepatitis A. One study reported "serological tests for viral infection" without reporting those tests, three others reported laboratory screening for "hepatitis" without providing further details (Table 3). Eighteen studies (23%) did not report any laboratory screening tests for enrollment (i.e., volunteers were enrolled if they or their caregivers reported good health); five of these 18 studies were in children and infants.

Most studies were cross-sectional; two studies (3%) were prospective, with follow up visits that compared resultant RIs over time, one in infants in Zimbabwe another in pregnant and postpartum women in Kenya (Table 3).

## Laboratory analytes tested

We found 77 different laboratory parameters across the 80 studies in this review (Table 1). 66 (83%) papers include hematology parameters, 23 (29%) included immunology parameters ranging from immunoglobulin and CD4+ T cell counts, to more in-depth delineation of T cell subsets (Tables 1 and 3), 32 (40%) papers included liver function parameters, and 30 (38%) included renal function parameters. The number of parameters characterized in each

manuscript ranged from only one (three studies each characterized either CD4+ counts, D-dimer, or hemoglobin), to as many as 40 (Table 3). A total of 54 different analysis platforms were reported, among the most common machines were Sysmex KX-21N (11), Beckton Dickson FACSCount (7) and FACSCalibur (8) cytometers, and Beckman Coulter ACT 5 Diff Hematology analyzer (9); two studies did not report their analysis platforms (Table 3). Statistical methods for calculating RIs varied; 46 manuscripts (58%) cited CLSI guidelines for defining RIs, five manuscripts (6%, all published since 2018) cited the newer C-RIDL guidelines for defining RIs, and 36 (45%) manuscripts described other methods, typically using nonparametric methods by reporting the observed 2.5th and 97.5th percentiles to describe a 95% RI (i.e., similar to the CLSI guidelines).

## Comparisons with RIs for other populations

While all studies noted the importance of regionally appropriate RIs, only 56 (70%) studies presented their results compared to other RIs in an effort to describe regional differences. Most (35/56, 63%) presented them side by side with limited analysis or discussion. The remainder (21, 37%) presented some information on the number and percentage of their study population that would be tallied as "out of range" against these comparator RIs, with the most common comparator range for this exercise being the US Division of AIDS (DAIDS) adverse events grading tables (n = 9). Four (7%) manuscripts presented statistical analysis to show that the number of out of range participants differed significantly from comparator ranges, though these statistical methods were often unclear; one additional manuscript tested whether or not male or female RIs fell 'out of range' more frequently, but did not test their generated RIs against comparator RIs. None reported perfect correlation with values in Europe or North America; only one study from South Africa with a high prevalence of 'mixed race' volunteers and individuals found that the hematological parameters measured did not vary meaningfully from international RIs and recommended ongoing use of the internationally derived RIs (Table 3).

## Discussion

Regionally appropriate reference intervals are a vital component to guiding the development, design, conduct and interpretation of clinical trials, as well as serving to support patient care. There is a paucity of published, rigorously conducted RI data available from sub-Saharan Africa. In this report, we summarize the diverse range of studies that describe laboratory reference intervals published after 2009 and before October 2021 (and conducted since 2006). We found significant regional heterogeneity; nearly one-third of all studies originated in Ethiopia, while central African countries were largely unrepresented. Many of the studies were at risk of bias that limited their suitability for use as reference intervals for future clinical trials; Most studies included significant issues including unrepresentative study populations, varying definitions of "healthy," and different age cut offs for population strata. Misunderstanding what is 'normal' in a population can lead to difficulties in clinical research. Selection of study populations and inclusion/exclusion criteria for clinical trials may be set in such a way as to inhibit enrolment or bias outcomes, if based on inappropriate RIs. Setting hemoglobin requirements that will be met by African men but not by most African women of reproductive age, for example, could skew a study population away from gender equality. Additionally, we observed that study populations were highly varied and selected populations underrepresented (e.g., the elderly, people living with HIV). Though methods to define RIs have been available during the period of this review, nearly half the studies do not cite the CLSI or C-RIDL guidelines. The parameters measured were diverse, as were the analysis platforms used to measure them.

There was near consensus across all studies that regionally appropriate RIs are important, but many investigators failed to present strong, consistent evidence for this. We recognize the limitations of searching published work for this type of data; Although we searched a diverse set of publication databases, some health care facilities may not have published their RIs, and we may have missed these data.

Since 1992, the Clinical and Laboratory Standards Institute (CLSI) in collaboration with the International Federation of Clinical Chemistry and Laboratory Medicine (IFCC) has published guidelines on the generation of new RIs. Now in its third edition [1], this document provides instruction for readers to create clinical laboratory RIs that "meet the minimum requirements for reliability and usefulness." The document lays out recommendations to select an appropriate study population, possible exclusion criteria to stimulate discussion of how to approach enrolling "healthy" participants, suggestions on partitioning criteria (e.g., age, sex, race, pregnancy / stage of pregnancy), and an example study questionnaire to modify as needed and administer to participants. Topics also include pre-analytical considerations (e.g., participant preparation including fasting, timing of sample collection, abstinence from certain drugs, etc.) and sample collection, handling, and testing. The report recommends 120 persons per partition to ensure adequate numbers to establish a RI using non-parametric methods. Discussion of methods to identify and exclude outliers in the data are included. Once outliers are removed, the lower limit of the RI is thus the 2.5th percentile of the data set, and the upper limit is the 97.5th percentile. All this should be done in the context of a robust quality control program, including equipment maintenance and EQA, to ensure precise and accurate data.

Increasingly common are large, multi-center studies for region specific RIs [12]. In concert with these studies, the IFCC Committee on Reference Intervals and Decision Limits (C-RIDL) set out to develop more robust guidelines for developing RIs than available via the above described CLSI document, and to explore the feasibility of conducting large scale studies to compare data across countries and regions. Initial efforts focused on three chemistry assays, ALT, AST and GGT, done in Turkey, China, Spain and across the Nordic countries; data were similar for ALT and AST, suggesting the use of common RIs were appropriate in some cases, but that was not the case for GGT [12]. Despite the obvious gap of data from Latin American and Africa, it was encouraging to see some harmonization was possible.

Building on this work, C-RIDL led a worldwide project to harmonize the generation of RIs, publishing a protocol and SOPs [3] to help others generate comparable data. A major change was to increase the recommended sample size from the CLSI-recommended 240 (i.e., 120 men and 120 women) to 500. This larger sample size allows the comparison of data across multiple sites and enables investigators to control for independent variables (e.g., smoking, diet, race, alcohol consumption) and to establish whether some RIs might be suitable for global standardization. Additional age-stratification among adults (e.g., 10-year age brackets among adults) was also recommended. Newer methods have been described to help identify outlier values (e.g., latent abnormal value exclusion, LAVE [13]) and to better describe when post-enrollment exclusion can be done. The newer C-RIDL protocol includes more guidance on subject selection, data and sample collection and management, standardization of assay results, additional statistical methods to generate the RIs, and greater detail on when to partition data.

As whole genome data becomes more available and inexpensive, a future avenue for RI research may include incorporating genetics as partitioning factor or an independent variable. Studies have shown the influence of genetic variation on laboratory RIs and the integration of these data into efforts to generate new RIs could become routine. Integrating genetic data into the generation of RIs should allow physicians and clinical trialists greater precision when interpreting individual laboratory parameter results [14, 15]. Lowering costs and more widespread technology means these data may become more available for African researchers [16].

Unfortunately, these data are more prevalent for European populations, further highlighting the importance of doing this work in underrepresented regions [17].

There remains a strong need to generate region-specific RIs to guide clinical trials and patient care in sub-Saharan Africa. Newer methods adopted globally are streamlining generation, comparison, and interpretation of these RIs [2, 3, 13]. Additional resources and studies to fill this gap are warranted, including a focus on improving awareness of newer methods.

## Supporting information

**S1 Checklist. PRISMA 2020 checklist.**
(DOCX)

**S1 Data. Original data used for review.**
(XLS)

## Acknowledgments

We thank Jackson Dykman at Explicom and Nicole Sender from IAVI for their help with Fig 2, the mapping of studies and populations. We thank Carl Verlinde for his early support on helping start this project.

## Author Contributions

**Conceptualization:** Matt A. Price, Patricia E. Fast, Mercy Mshai, Maureen Lambrick, Lisa Gieber, Paramesh Chetty, Vincent Muturi-Kioi.

**Data curation:** Matt A. Price, Patricia E. Fast, Maureen Lambrick, Lisa Gieber, Paramesh Chetty, Vincent Muturi-Kioi.

**Formal analysis:** Matt A. Price, Patricia E. Fast, Yvonne Wangũi Machira, Vincent Muturi-Kioi.

**Funding acquisition:** Matt A. Price.

**Investigation:** Matt A. Price.

**Methodology:** Matt A. Price, Patricia E. Fast, Mercy Mshai, Yvonne Wangũi Machira, Lisa Gieber, Paramesh Chetty, Vincent Muturi-Kioi.

**Project administration:** Matt A. Price, Maureen Lambrick, Lisa Gieber.

**Resources:** Yvonne Wangũi Machira, Lisa Gieber, Paramesh Chetty.

**Supervision:** Matt A. Price.

**Visualization:** Matt A. Price.

**Writing – original draft:** Matt A. Price.

**Writing – review & editing:** Matt A. Price, Patricia E. Fast, Mercy Mshai, Maureen Lambrick, Yvonne Wangũi Machira, Lisa Gieber, Paramesh Chetty, Vincent Muturi-Kioi.

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
