## [Decision Letter · Decision Letter 0]

23 May 2022

PGPH-D-22-00497

Region-specific laboratory reference intervals are important: a systematic review of the data from Africa

Dear Dr. Price,

Thank you for submitting your manuscript to PLOS Global Public Health. After careful consideration, we feel that it has merit but does not fully meet PLOS Global Public Health’s publication criteria as it currently stands. Therefore, we invite you to submit a revised version of the manuscript that addresses the points raised during the review process.

Please submit your revised manuscript by . If you will need more time than this to complete your revisions, please reply to this message or contact the journal office at globalpubhealth@plos.org. Please include the following items when submitting your revised manuscript:

We look forward to receiving your revised manuscript.

Kind regards,

Nikita Mehra, M.D., DM.,

Academic Editor

Journal Requirements:

supported your study, including funding received from your institution. 

- State the initials, alongside each funding source, of each author to receive each grant.

2. Please provide separate figure files in .tif or .eps format.

3. We do not publish any copyright or trademark symbols that usually accompany proprietary names, eg (R), (C), or TM  (e.g. next to drug or reagent names). Please remove all instances of trademark/copyright symbols throughout the text, including ®, TM on pages 43 and 31.

4. Figure 2: please (a) provide a direct link to the base layer of the map used and ensure this is also included in the figure legend; (b) provide a link to the terms of use / license information for the base layer. We cannot publish proprietary or copyrighted maps (e.g. Google Maps, Mapquest) and the terms of use for your map base layer must be compatible with our CC-BY 4.0 license. 

5. We have noticed that you have uploaded Supporting Information files, but you have not included a list of legends. Please add a full list of legends for your Supporting Information files after the references list. 

Additional Editor Comments (if provided):

Thank you for providing a descriptive analysis of your work. We look forward to your resubmission of a revised manuscript based on comments and suggestions from the reviewers. We look forward to receiving a revised manuscript with policymaking implications in addition to the descriptive analysis.

Reviewers' comments:

Reviewer's Responses to Questions

**Comments to the Author**

1. Does this manuscript meet PLOS Global Public Health’s publication criteria? Is the manuscript technically sound, and do the data support the conclusions? The manuscript must describe methodologically and ethically rigorous research with conclusions that are appropriately drawn based on the data presented.

Reviewer #1: Partly

Reviewer #2: Yes

2. Has the statistical analysis been performed appropriately and rigorously?

Reviewer #1: No

Reviewer #2: N/A

3. Have the authors made all data underlying the findings in their manuscript fully available (please refer to the Data Availability Statement at the start of the manuscript PDF file)?

Reviewer #1: Yes

Reviewer #2: Yes

4. Is the manuscript presented in an intelligible fashion and written in standard English?

Reviewer #1: No

Reviewer #2: Yes

5. Review Comments to the Author

Reviewer #1: 1. The word "summary" not necessary to say please change to "Abstract "

2. There are a lot of work "we" through out in the document please remove it

3. Location of abbreviation list inappropriate

4. Discussion section totally not like discussion just description of the your finding please rewrite it

5. General the manuscript not followed this journal guidelines

Reviewer #2: The authors showed genuine effort in their methodology to achieve the objective of the study. They also showed results that is relevant to scientific and clinical practice. However I find the conclusion weak in recommending the application of newer and standardized methods of RI generation studies, though this was mentioned.

Are there reasons for limiting the literature search to work done after 2006? Considering that this study is on reference intervals for analytes in clinical diagnosis and trials, less emphasis on 'focusing on recent work' may have improved robustness of findings. Kindly clarify this choice in the method section.

6. PLOS authors have the option to publish the peer review history of their article (what does this mean?). If published, this will include your full peer review and any attached files.

**Do you want your identity to be public for this peer review?** For information about this choice, including consent withdrawal, please see our Privacy Policy.

Reviewer #1: No

Reviewer #2: No

---

## [Decision Letter · Decision Letter 1]

6 Oct 2022

Region-specific laboratory reference intervals are important: a systematic review of the data from Africa

PGPH-D-22-00497R1

Dear Dr. Price,

We are pleased to inform you that your manuscript 'Region-specific laboratory reference intervals are important: a systematic review of the data from Africa' has been provisionally accepted for publication in PLOS Global Public Health.

Best regards,

Nikita Mehra, M.D., DM.,

Academic Editor

Reviewer Comments (if any, and for reference):

Reviewer's Responses to Questions

**Comments to the Author**

1. If the authors have adequately addressed your comments raised in a previous round of review and you feel that this manuscript is now acceptable for publication, you may indicate that here to bypass the “Comments to the Author” section, enter your conflict of interest statement in the “Confidential to Editor” section, and submit your "Accept" recommendation.

Reviewer #1: All comments have been addressed

2. Does this manuscript meet PLOS Global Public Health’s publication criteria? Is the manuscript technically sound, and do the data support the conclusions? The manuscript must describe methodologically and ethically rigorous research with conclusions that are appropriately drawn based on the data presented.

Reviewer #1: Partly

3. Has the statistical analysis been performed appropriately and rigorously?

Reviewer #1: Yes

4. Have the authors made all data underlying the findings in their manuscript fully available (please refer to the Data Availability Statement at the start of the manuscript PDF file)?

Reviewer #1: Yes

5. Is the manuscript presented in an intelligible fashion and written in standard English?

Reviewer #1: Yes

6. Review Comments to the Author

Reviewer #1: None
